



**Significant production of ClNO$_2$ and possible source of Cl$_2$ from N$_2$O$_5$ uptake at a**
**suburban site in eastern China**
Men Xia[a], Xiang Peng[a], Weihao Wang[a], Chuan Yu[a], Peng Sun[b], Yuanyuan Li[b], Yuliang Liu[b],
Zhengning Xu[b], Zhe Wang[a, c], Zheng Xu[b], Wei Nie[b], Aijun Ding[b], and Tao Wang[a,*]
[a] Department of Civil and Environmental Engineering, The Hong Kong Polytechnic University,
Hong Kong, China
[b] Joint International Research Laboratory of Atmospheric and Earth System Sciences, School
of Atmospheric Sciences, Nanjing University, Nanjing, 210023, China
[c] Now at Division of Environment and Sustainability, Hong Kong University of Science and
Technology, Hong Kong, China
*Correspondence to*: Tao Wang (cetwang@polyu.edu.hk)
**Abstract**
ClNO$_2$ and Cl$_2$ can affect atmospheric oxidation and thereby the formation of ozone and
secondary aerosols, yet their sources and production mechanisms are not well understood or
quantified. In this study we present field observations of ClNO$_2$ and Cl$_2$ at a suburban site in
eastern China during April 2018. Persistent high levels of ClNO$_2$ (maximum ~3.7 ppbv; 1 min
average) were frequently observed at night, due to the high ClNO$_2$ yield ($\varphi$(ClNO$_2$), $0.56 \pm 0.20$)
inferred from the measurements. The $\varphi$(ClNO$_2$) value showed a positive correlation with the
[Cl$^-$]/[H$_2$O] ratio, and its parameterization was improved by the incorporation of [Cl$^-$]/[H$_2$O]
and the suppression effect of aerosol organics. ClNO$_2$ and Cl$_2$ showed a significant correlation
on most nights. We show that the Cl$_2$ at our site was likely a co-product with ClNO$_2$ from N$_2$O$_5$
uptake on aerosols that contain acidic chloride, rather than being produced by ClNO$_2$ uptake,
as previously suggested. The Cl$_2$ yield ($\varphi$(Cl$_2$)) derived from the N$_2$O$_5$ uptake hypothesis
exhibited significant correlations with [Cl$^-$] and [H$^+$], based on which a parameterization of
$\varphi$(Cl$_2$) was developed. The derived parameterizations of $\varphi$(ClNO$_2$) and $\varphi$(Cl$_2$) can be used in
models to quantify the nighttime production of ClNO$_2$ and Cl$_2$ and their impact on the next
day's photochemistry.
**Graphical abstract**

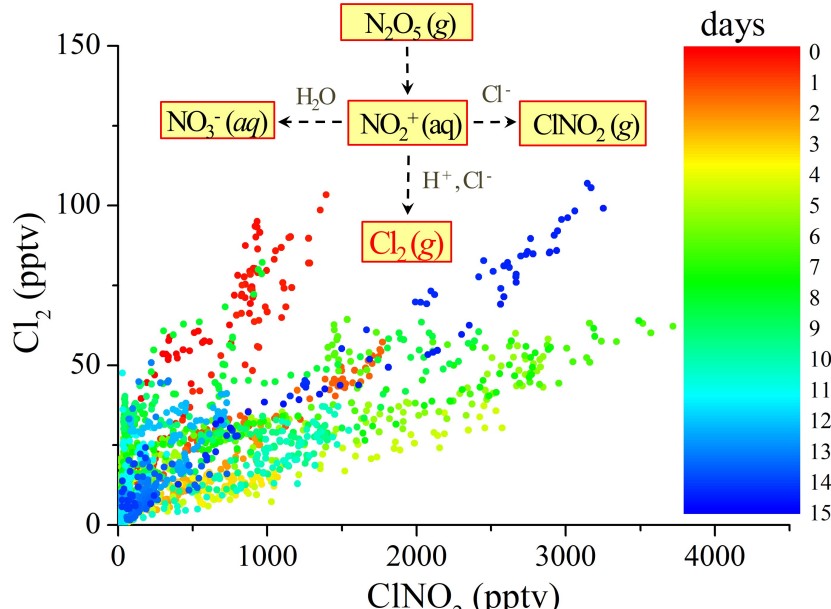


**1. Introduction**
Chlorine radicals (Cl·) are potent oxidizers in the atmosphere (Seinfeld and Pandis, 2016).
Cl· destroy the $O_3$ layer in the stratosphere, exposing the biosphere to excess ultraviolet
radiation (Molina and Rowland, 1974). In the polluted troposphere, Cl· react with volatile
organic compounds (VOCs), especially alkanes, contribute to primary $RO_x$ (= $OH+HO_2+RO_2$)
production, and affect hydroxyl radical (OH) and $O_3$ concentrations (Simpson et al., 2015).
Nitryl chloride ($ClNO_2$) is a major chlorine radical precursor in the troposphere and has been
investigated around the globe over the past decade (Osthoff et al., 2008; Thornton et al., 2010;
Mielke et al., 2011; Wang et al., 2016). $ClNO_2$ is an important nocturnal reservoir of chlorine
and $NO_x$ and is produced mostly at night. $NO_x$ reacts with $O_3$ to form $NO_3$ radicals and $N_2O_5$
(Reactions R1 and R2). When aerosol chloride is present, $ClNO_2$ and nitrate are produced from
the heterogeneous uptake of $N_2O_5$ on aerosols (Reaction R3) (Finlayson-Pitts et al., 1989). After
sunrise, $ClNO_2$ is photolyzed to return $NO_2$ and release Cl· (Reaction R4).
$NO_2(g) + O_3(g) \rightarrow NO_3(g) + O_2(g)$ (R1)
$NO_3(g) + NO_2(g) \leftrightarrow N_2O_5(g)$ (R2)



$N_2O_5(g) + Cl^-(aq) \rightarrow ClNO_2(g) + NO_3^-(aq)$     (R3)
$ClNO_2(g) + h\nu \rightarrow Cl\cdot(g) + NO_2(g)$     (R4)
Two key kinetic parameters for quantification of $ClNO_2$ formation are $\gamma(N_2O_5)$ (i.e., $N_2O_5$
uptake probability on aerosols) and $\varphi(ClNO_2)$ (i.e., $ClNO_2$ production yield from $N_2O_5$ uptake)
(Thornton et al., 2003; Behnke et al., 1997). Laboratory studies have shown that $\varphi(ClNO_2)$ is
dependent on the $[Cl^-]/[H_2O]$ ratio because aqueous $Cl^-$ and $H_2O$ compete for the $NO_2^+$
intermediate, based upon which a parameterization was developed to predict $\varphi(ClNO_2)$
(hereafter denoted as $\varphi(ClNO_2)_{BT}$) (Bertram and Thornton, 2009). The parameterization was
tested in several field studies, and it was found that the parameterized $\varphi(ClNO_2)$ values were
significantly larger than the field-derived values (Tham et al., 2016; Wang et al., 2017; Tham
et al., 2018; McDuffie et al., 2018b; Staudt et al., 2019). The exact causes of these discrepancies
are not fully understood. The suppression of $\varphi(ClNO_2)$ has been observed in biomass-burning
plumes in north China, but the specific species that reduced $\varphi(ClNO_2)$ were not identified
(Tham et al., 2018). Some inorganic nucleophiles such as sulfate and organic nucleophiles such
as acetate were recently proposed to decrease $\varphi(ClNO_2)$ by consuming $NO_2^+$ (McDuffie et al.,
2018b; Staudt et al., 2019). Such $NO_2^+$-consuming nucleophiles may generate products from
$N_2O_5$ uptake other than $ClNO_2$ and nitrate, and this is deserving of further investigation.
Besides $ClNO_2$, $Cl_2$ is another important chlorine radical precursor that is present in the lower
troposphere (Spicer et al., 1998; Custard et al., 2016; Priestley et al., 2018). Elevated levels of
$Cl_2$ (up to ~400 pptv) have been observed during the daytime in polar and continental
environments (Liao et al., 2014; Liu et al., 2017), whereas other studies found nocturnal peaks
of $Cl_2$ mixing ratios in polar, coastal, and continental sites (Mielke et al., 2011; Riedel et al.,
2012; Riedel et al., 2013; McNamara et al., 2019). Several potential sources of $Cl_2$ have been
proposed, such as direct emissions from power plants (Riedel et al., 2013) and water treatment
facilities (Mielke et al., 2011), photochemical formation associated with $O_3$ (Liao et al., 2014),
and heterogeneous conversion from chlorinated compounds (Reactions R5 and R6) (Deiber et
al., 2004; Pratte and Rossi, 2006; McNamara et al., 2019).
$HOCl(g) + H^+(aq) + Cl^-(aq) \rightarrow Cl_2(g) + H_2O$     (R5)



$ClONO_2(g) + H^+(aq) + Cl^-(aq) \rightarrow Cl_2(g) + HNO_3(aq)$        (R6)
$Cl_2$ can also be produced from heterogeneous $N_2O_5$ uptake on acidic aerosols laden with
chloride, and $ClNO_2(aq)$ has been proposed as an intermediate in $Cl_2$ production (Reaction R7)
on the basis of laboratory studies (Roberts et al., 2008; Roberts et al., 2009).
$ClNO_2(aq) + H^+(aq) + Cl^-(aq) \rightarrow Cl_2(g) + HNO_2(aq)$        (R7)
Significant correlations of $ClNO_2$ and $Cl_2$ were observed during an airborne campaign in the
United States and were interpreted as evidence of $Cl_2$ production from $ClNO_2$ uptake on acidic
aerosols (Haskins et al., 2019). However, this study also found that $Cl_2$ formation from $ClNO_2$
uptake was less efficient, because the estimated $\gamma(ClNO_2)$ value ($(2.3 \pm 1.8) \times 10^{-5}$) was two
orders of magnitude lower than that suggested by laboratory studies ($(6.0 \pm 2.0) \times 10^{-3}$) (Roberts
et al., 2008; Haskins et al., 2019). It remains unclear whether $ClNO_2$ uptake proceeds more
slowly in ambient environments than in laboratory conditions or whether additional pathways
are responsible for the formation of $Cl_2$. Therefore, the detailed activation process by which
inert chlorine (e.g., particulate chloride) is converted to reactive chlorine remains highly
uncertain and requires further research.
In April 2018, we conducted field measurements of $ClNO_2$, $Cl_2$, and other trace gases and
aerosols in a suburban area of the Yangtze River Delta (YRD), a highly populated and
industrialized region in eastern China. High levels of $ClNO_2$ with enhanced $Cl_2$ were observed
at night. In this study, we investigated the activation of chlorine initiated by heterogeneous $N_2O_5$
chemistry. We first introduce prominent features of the observation results. The key parameters
in $ClNO_2$ formation (i.e., $\gamma(N_2O_5)$ and $\varphi(ClNO_2)$) are then derived using the ambient data.
Factors that influence $\varphi(ClNO_2)$ are discussed, with a focus on a revision of the
parameterization of $\varphi(ClNO_2)$. We present observational evidence for a possible co-production
pathway of $Cl_2$ with $ClNO_2$ from heterogeneous reactions of $N_2O_5$ and propose a new
parameterization for nocturnal formation of $Cl_2$.

**2. Methods**
**2.1 Observation sites**



The field campaign was conducted from 11 to 26 April, 2018 on the Xianlin Campus of
Nanjing University, which is situated in a suburban area approximately 20 km northeast of
downtown Nanjing (see Fig. 1). The observation sites are surrounded by teaching and
residential buildings, sparse roads, and vegetation cover for about 1 to 2 km, with no significant
emission sources. Approximately 15 km northwest of the sampling sites are large-scale
chemical and steel facilities, which can be sources of gaseous pollutants (CO, $SO_2$, $NO_x$, and
VOCs) and particulate matters that may influence the site (Zhou et al., 2017). In addition,
Shanghai is approximately 270 km southeast of the measurement site.
The main data reported in this study (i.e., $N_2O_5$, $ClNO_2$, and $Cl_2$) and the $NO_x$ and $O_3$ data
were obtained at the School of Atmospheric Sciences (SAS) of Nanjing University (sampling
site 1). The auxiliary data, including $O_3$, VOCs, aerosol size distribution, and chemical
composition, were obtained at the Station for Observing Regional Processes of the Earth
System (SORPES, sampling site 2). Fig. 1 shows the locations of the two sampling sites.
Interested readers are referred to previous studies for more information about the SORPES site
(e.g., Ding et al., 2013; Sun et al., 2018; Ding et al., 2019). A comparison of $O_3$ measurements
at the SAS and SORPES sites shows excellent agreement during the observation period (Fig.
S1).



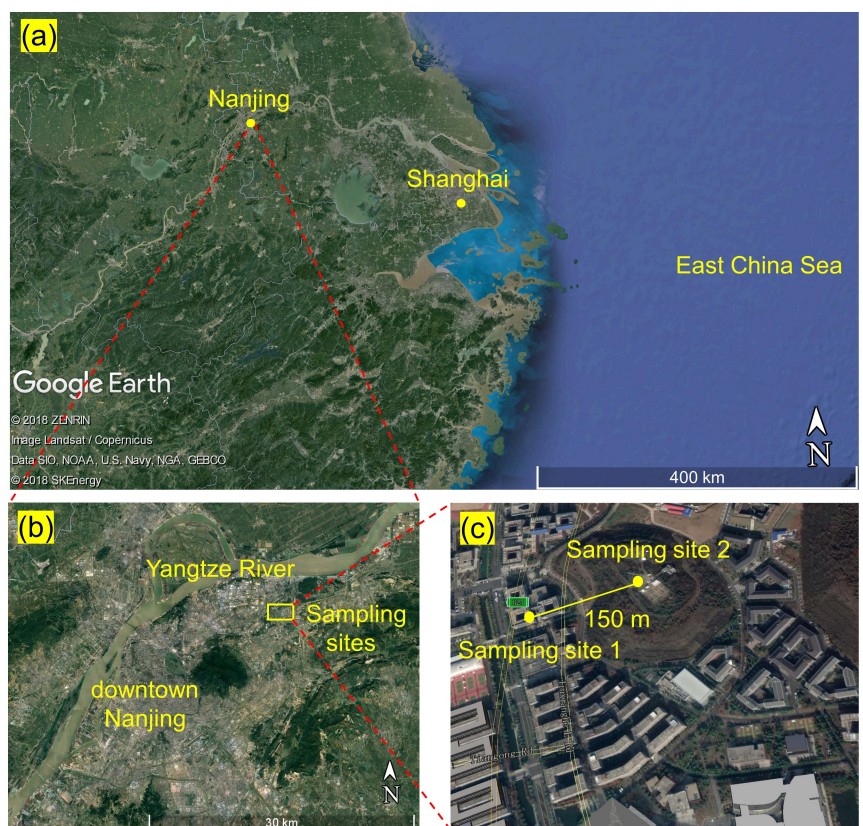


**Figure 1.** Sampling locations. **(a)** Location of Nanjing city in the YRD region. **(b)** Location of
sampling sites in Nanjing. **(c)** Sampling sites 1 and 2 on the Xianlin campus of Nanjing
University. (© Google Earth)

**2.2 N₂O₅, ClNO₂, and Cl₂ measurements**
A chemical ionization mass spectrometer coupled with a quadrupole mass analyzer (Q-
CIMS, THS Instruments) was used to detect $N_2O_5$, $ClNO_2$, $Cl_2$, and $HOCl$. The Q-CIMS had
been used in previous field campaigns to measure $N_2O_5$ and $ClNO_2$ (Wang et al., 2016; Tham
et al., 2016). In this study, we also measured $Cl_2$ and $HOCl$ and tuned the pressure of the drift
tube reactor accordingly. The principles and ion chemistry of Q-CIMS were described in detail
by Kercher et al. (2009). Briefly, iodide ($I^-$) was adopted as the primary ion for strong affinity
with our target species. Charged iodide clusters, such as $IN_2O_5^-$, $IClNO_2^-$, $ICl_2^-$, and $IHOCl^-$, are





formed by the ion molecular reactions shown in Reactions (R8) through (R11). Ion clusters with
different Cl isotopes (i.e., $^{35}$Cl and $^{37}$Cl) were recorded to examine the identity of $ClNO_2$ and
$Cl_2$, and this isotopic analysis confirmed that $ClNO_2$ and $Cl_2$ had very minor interferences (see
Text S1).
$N_2O_5 + I^- \rightarrow IN_2O_5^-$ (*m/z* 235)                                       (R8)
$ClNO_2 + I^- \rightarrow IClNO_2^-$ (*m/z* 208, 210)                              (R9)
$Cl_2 + I^- \rightarrow ICl_2^-$ (*m/z* 197, 199)                                 (R10)
$HOCl + I^- \rightarrow IHOCl^-$ (*m/z* 179, 181)                            (R11)
The Q-CIMS was housed on the fifth floor of the SAS building. The PFA sampling tube
(length, 1.5 m; outer diameter, 0.25 in) extended out through a hole in the side wall. We took
precautions to minimize the deposition of particles on the inner wall of the sampling tube and
tested the possible formation and loss of $N_2O_5$, $ClNO_2$, and $Cl_2$ on the sampling tube (see Text
S1 for details), which showed a negligible inlet interference on the CIMS measurement. $N_2O_5$
and $ClNO_2$ were calibrated every two days following established methods (Wang et al., 2016).
Briefly, $N_2O_5$ was synthesized from the reaction of $NO_2$ and $O_3$, and $ClNO_2$ was produced by
passing $N_2O_5$ through a deliquesced NaCl slurry. The dependence of $N_2O_5$ sensitivity on relative
humidity (RH) was tested on site (see Fig. S3) and was used to account for changes in ambient
RH. A $Cl_2$ permeation tube was used for $Cl_2$ calibration (Liao et al., 2014), and the permeation
rate of $Cl_2$ was quantified by chemical titration and ultraviolet spectrophotometry. We assumed
the sensitivity of HOCl to be the same as that of ClO, and we used a sensitivity ratio of ClO to
$Cl_2$ (0.26) that was experimentally determined by Custard et al. (2016). In this study, the HOCl
data were only used qualitatively. In sum, the sensitivities of $N_2O_5$, $ClNO_2$, $Cl_2$, and HOCl were
$0.42 \pm 0.11$, $0.35 \pm 0.13$, $0.86 \pm 0.37$, and $0.22 \pm 0.08$ Hz/pptv, respectively. The uncertainties
of the $N_2O_5$ and $ClNO_2$ measurements were estimated to be 19% via error propagation. The $Cl_2$
measurement uncertainty was estimated to be 15%. The details of CIMS calibrations and
uncertainty analysis are available in Text S1 and Table S3.

**2.3 Auxiliary measurements**





In addition to the CIMS measurement at the SAS site, meteorological factors, gaseous and
aerosol chemical compositions, particle size distributions, and the $NO_2$ photolysis frequency
($jNO_2$) were simultaneously measured at the SORPES site (Table S1). The ionic compositions
of $PM_{2.5}$, including $Cl^-$, $NO_3^-$, $SO_4^{2-}$, and $NH_4^+$, were measured with an Aerosol Chemical
Speciation Monitor (ACSM, Aerodyne Research Inc.) and MARGA (Metrohm, Switzerland).
The hourly-averaged ionic compositions from ACSM and MARGA showed good agreement
(see Fig. S4). In addition, $HNO_3$ was also measured by MARGA. In this study, the 10-min
averaged ACSM data, including total organics, were used for subsequent analysis. The mass
concentration of $H^+$ ($\mu g/m^3$) was estimated to achieve electric charge balance of the cation
($NH_4^+$) and anions ($Cl^-$, $NO_3^-$, and $SO_4^{2-}$) of the ACSM data. The molar concentrations of
inorganic ions (i.e., $[Cl^-]$, $[NO_3^-]$, $[SO_4^{2-}]$, $[NH_4^+]$, and $[H^+]$) and total organics ($[Org]$) were
estimated using the extended aerosol inorganics model (E-AIM, model III) (Wexler, 2002). The
molecular weight of the organic molecules was assumed to be 250 g/mol (McDuffie et al.,
2018b). The dry-state submicron particle size distribution was measured with a Scanning
Mobility Particle Sizer (SMPS, TSI Inc.), and the data were used to estimate the aerosol surface
area density ($S_a$) with the assumption of spherical particles. The hygroscopic growth factor of
the particle size was based on an empirical parameterization, $GF = 0.582 \left( 8.46 + \dfrac{1}{1 - RH} \right)^{1/3}$
(Lewis, 2008). The VOCs were measured with a proton transfer reaction time-of-flight mass
spectrometer (PTR-TOF-MS, Ionicon).

**2.4 Production and loss of $NO_3$ and $N_2O_5$**
$NO_3$ radicals are primarily produced from $NO_2$ and $O_3$ (Reaction R1). The production rate
equation of $NO_3$ ($P(NO_3)$) is shown as follows (Eq. (1):
$P(NO_3) = k_1[NO_2][O_3]$                 (1)
where $k_1$ is the rate constant of Reaction R1. $NO_3$ is mainly removed by gas-phase reactions
with VOCs and NO (Eq. (2)) and heterogeneous loss via $N_2O_5$ uptake (Eq. (3)), where $k(NO_3)$
and $k(N_2O_5)$ are the first-order loss rate coefficients of $NO_3$ and $N_2O_5$, respectively.
$k(NO_3) = k_{NO + NO_3}[NO] + \sum k_i[VOC_i]$              (2)


$k(N_2O_5) = \frac{1}{4}c(N_2O_5)S_a\gamma(N_2O_5)$ (3)
where $k_{NO+NO_3}$ and $k_i$ denote the reaction rate constants of $NO_3$ with NO and VOC,
respectively, and $c(N_2O_5)$ is the average velocity of $N_2O_5$ molecules. Other minor loss pathways
of $NO_3$ and $N_2O_5$ were not considered (e.g., homogeneous loss of $N_2O_5$).

**2.5 Estimation of $\varphi(ClNO_2)$ and $\gamma(N_2O_5)$**
$\varphi(ClNO_2)$ and $\gamma(N_2O_5)$ were estimated using the observation data and parameterization. We
used the observed increasing rates of $ClNO_2$ and total nitrate (i.e., $HNO_3+NO_3^-$) to derive the
values for $\gamma(N_2O_5)$ and $\varphi(ClNO_2)$ in the selected cases (Phillips et al., 2016). Details of the
method are described elsewhere (Tham et al., 2016; Phillips et al., 2016). Briefly, the production
rate of $ClNO_2$ ($P(ClNO_2)$) is calculated as follows (Eq. (4)).
$P(ClNO_2) = \frac{1}{4}c(N_2O_5)S_a\gamma(N_2O_5)[N_2O_5]\varphi(ClNO_2)$ (4)
The production rate of total nitrate induced by $N_2O_5$ uptake during the night ($P(NO_3^-)$) is shown
by Eq. (5).
$P(NO_3^-) = \frac{1}{4}c(N_2O_5)\,S_a\gamma(N_2O_5)[N_2O_5](2-\varphi(ClNO_2))$ (5)
$\varphi(ClNO_2)$ is obtained by combining Eqs. (4) and (5).
$\varphi(ClNO_2) = 2(1+\dfrac{P(NO_3^-)}{P(ClNO_2)})^{-1}$ (6)
And $\gamma(N_2O_5)$ is derived as follows (Eq. (7)).
$\gamma(N_2O_5) = \dfrac{2(P(ClNO_2) + P(NO_3^-))}{c(N_2O_5)S_a[N_2O_5]}$ (7)
This method assumes that: (1) air masses are relatively stable; and (2) $N_2O_5$ uptake dominates
$NO_3^-$ production at night (Tham et al., 2018). Assumption (1) requires careful selection of the
cases of interest. Regarding assumption (2), major nocturnal production pathways of total
nitrate should be evaluated, such as comparing the reaction rate of $N_2O_5$ heterogeneous loss
($k(N_2O_5)*[N_2O_5]$) with that of $NO_3 + VOC$ ($k(NO_3)*[NO_3]$), which may produce $HNO_3$ via H-
abstraction reactions.
$\varphi(ClNO_2)$ was also calculated with the parameterization shown in Eq. (8), in which the $k_4/k_3$
ratio was adopted as $483 \pm 175$ (Bertram and Thornton, 2009).





$$\varphi(ClNO_2)_{BT} = \left(1 + \frac{[H_2O]}{k_4/k_3[Cl^-]}\right)^{-1} \tag{8}$$
When considering the potential competitive effect of other species (denoted as "Y⁻"), such as
sulfate or aerosol organics, for the $NO_2^+$ intermediate, the following equation (Eq. (9)) was
established (McDuffie et al., 2018b). Rearrangement of Eq. (9) yields Eq. (10), in which
plotting $\left(\frac{1}{\varphi(ClNO_2)} - 1\right) * \frac{[Cl^-]}{[H_2O]}$ to $\frac{[Y^-]}{[Cl^-]}$ should exhibit a positive correlation. $k_5$ represents a
constant reaction rate coefficient of "Y⁻" with $NO_2^+$.
$$\varphi(ClNO_2) = \frac{1}{1 + \frac{k_3[H_2O]}{k_4[Cl^-]} + \frac{k_5[Y^-]}{k_4[Cl^-]}} \tag{9}$$
$$\left(\frac{1}{\varphi(ClNO_2)} - 1\right) * \frac{[Cl^-]}{[H_2O]} = \frac{k_3}{k_4} + \frac{k_5[Y^-]}{k_4[Cl^-]} \tag{10}$$

**3. Results and Discussions**
**3.1 Overall observation results**
Fig. 2 depicts the time series of $N_2O_5$, $ClNO_2$, $Cl_2$, and related species. Overall, the
observation sites experienced moderate levels of pollution during the study period ($PM_{2.5}$, 44.8
± 18.3 μg/m³; CO, 0.4 ± 0.2 ppmv; $SO_2$, 3.1±1.8 ppbv; $NO_x$, 18.1 ± 16.6 ppbv; $O_3$, 25.8 ± 18.4
ppbv). The on-site observations indicated mostly stagnant weather with low wind speeds (1 m/s
in average). No precipitation was observed except for the evening of 13 April from 22:00 to
22:30 local time. The nocturnal NO mixing ratios were usually near the detection limit of the
NO instrument, and the presence of abundant $NO_2$ and $O_3$ favored $N_2O_5$ formation and
subsequent heterogeneous processes.
The most salient features of the observation were the high levels of $ClNO_2$ and moderate
levels of $Cl_2$ that were present during the night. The $ClNO_2$ mixing ratios exceeded 1 ppbv on
12 of the 15 nights. The observed $ClNO_2$ levels were among the highest in the world, with a
peak mixing ratio (1-min average, 3.7 ppbv) slightly higher than that of north China (1-min
average, 2.1 ppbv) (Tham et al., 2016) but lower than that reported in south China (1 min
average, 8.3 ppbv) (Yun et al., 2018). The frequent occurrence of high $ClNO_2$ levels was favored
by several factors, including elevated levels of $N_2O_5$ (1 ppbv), humid weather (RH, 67.7 ±
20.7%), and chloride availability ($0.36 \pm 0.31$ μg/m$^3$) during the field campaign. When high
levels of ClNO$_2$ were observed, elevated concentrations of particulate nitrate as high as 40.8
μg/m$^3$ (10-min average) were also present. We noticed that ClNO$_2$ and particulate nitrate
concentrations both increased more rapidly after midnight than before midnight from 15 to 19
April, which is discussed further below.
Moderate levels of Cl$_2$ (up to 100 pptv) were also observed during the night. Cl$_2$ mixing ratios
exhibited a clear diurnal pattern, peaking at night and decreasing during the day due to
photolysis. The nocturnal peaks of Cl$_2$ mixing ratios showed discrepancies from some previous
observations in which an elevated levels of Cl$_2$ was found during the day (Liao et al., 2014; Liu
et al., 2017). The Cl$_2$ and ClNO$_2$ mixing ratios reached peaks synchronously during most nights,
and both species decreased in abundance or were absent in NO-rich plumes (e.g., the nights of
13 and 25 April), which suggests that Cl$_2$ and ClNO$_2$ were produced from common sources.
Similar nighttime correlations of Cl$_2$ and ClNO$_2$ were also observed in the United States and in
northern China (Qiu et al., 2019; Haskins et al., 2019). A subsequent analysis of the present
study aims to elucidate the nighttime formation processes of ClNO$_2$ and Cl$_2$.

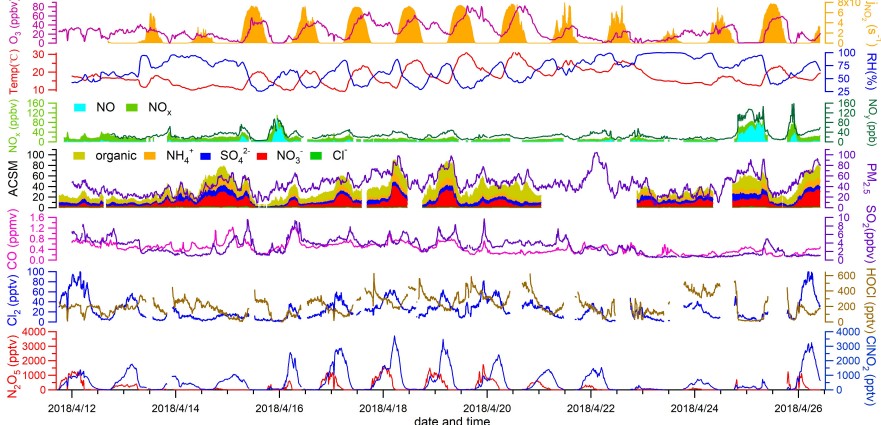


**Figure 2.** Time series of ClNO$_2$, Cl$_2$, and related measurements during field observations from
11 to 26 April 2018. Data gaps were caused by technical problems or calibrations.

**3.2 High ClNO$_2$ cases**
Fig. 3 shows the observation results from 17 and 18 April to further illustrate the ClNO$_2$





formation process. This case had the highest $ClNO_2$ observed during the campaign and shows
an example of high $ClNO_2$ mixing ratios after midnight. As shown in Fig. 3a, the mixing ratio
of $ClNO_2$ began to increase after sunset (18:00 17 April) and decreased after midnight. The
period between 22:00 and 24:00 on 17 April was noted as plume 1. After midnight, the $ClNO_2$
mixing ratios exhibited a more rapid increase from 03:00 to 05:00 on 18 April (plume 3), and
the particulate nitrate concentration also synchronously and significantly increased. Plumes 1
and 3 were identified as being different, resulting from an air mass shift between 00:00 and
03:00 on 18 April (plume 2), as indicated by abrupt changes in the RH, temperature, and $O_3$.
We compared the backward trajectories from plume 1 to plume 3 and found no significant
difference (figures not shown here). Thus, the change in the air mass from plume 1 to plume 3
was likely a local phenomenon.
The $P(NO_3)$ and $NO_3$ loss pathways during plumes 1 and 3 were calculated and compared in
Fig. 3b-d using the methods described in Section 2.4. The $P(NO_3)$ was slightly lower during
plume 3 than during plume 1, and a larger proportion of $NO_3$ was lost via the $N_2O_5$ hydrolysis
pathway in plume 3. Thus, the air mass shift, in addition to the higher rate of $N_2O_5$ hydrolysis,
was responsible for the elevated $ClNO_2$ levels observed after midnight.

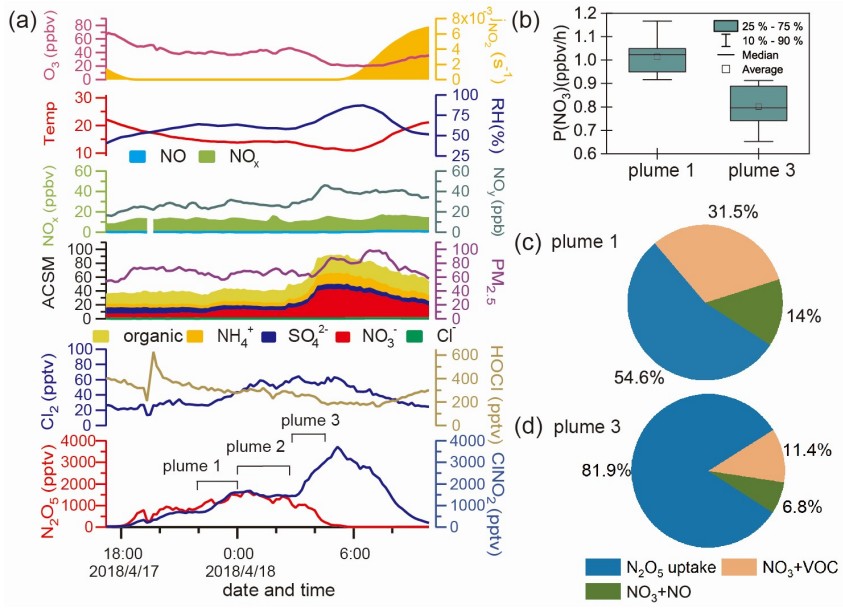




**Figure 3.** Detailed analysis of a high $ClNO_2$ episode observed on 17–18 April. **(a)** Time series
of $ClNO_2$ and related species. **(b)**, **(c)**, and **(d)** Comparisons of $P(NO_3)$ and $NO_3$ loss pathways
in plumes 1 and 3.

**3.3 $ClNO_2$ production yield from $N_2O_5$ uptake**
$\varphi(ClNO_2)$ was estimated to investigate its influencing factors and the performance of
parameterization in selected cases. The methods described in Section 2.5 were used to estimate
the $\varphi(ClNO_2)$ and $\gamma(N_2O_5)$ using the observation data. As these methods assume a stable air
mass and the dominance of $N_2O_5$ uptake in nitrate formation, we applied the following criteria
when selecting cases for this analysis. First, the NO mixing ratios must be less than 0.1 ppbv.
When significant levels of NO were present, the $N_2O_5$ chemistry was suppressed. Second,
primary pollutants such as CO, $SO_2$, and meteorological factors (wind, temperature, and RH)
were required to exhibit relatively constant levels or stable trends within the cases. Third, the
$ClNO_2$ and nitrate levels had to be correlated ($R^2 > 0.6$) and show increasing trends. Fifteen
cases that lasted 30 min to 3 hours were selected, and 10-min averaged data were used for
calculation. Fig. S5 shows an example of this calculation, which corresponds to plume 1 on 17
April (Fig. 3). We then evaluated the loss pathways of $NO_3$ in the fifteen cases. The results show
that the $NO_3$ + VOCs reactions contributed less than one third of the total $NO_3$ + $N_2O_5$ loss (e.g.,
Fig. 3c, d). Nocturnal total nitrate production was thus dominated by $N_2O_5$ uptake, and only a
small proportion of nitrate was produced by $NO_3$+VOCs reactions.
The derived $\gamma(N_2O_5)$ values ranged from 0.004 to 0.014 (mean, 0.008 ± 0.004). The highest
$\gamma(N_2O_5)$ values (0.0135 and 0.0139) were derived between 03:00 and 05:00 on 18 April (i.e.,
plume 3 in Fig. 3), which was consistent with the rapid increase in $ClNO_2$ mixing ratios during
that period. The variations in the $\gamma(N_2O_5)$ value depended mainly on $[H_2O]$ ($R^2 = 0.49$) (see Fig.
S6) but showed little correlation with other influencing factors, such as $[Cl^-]$, $[NO_3^-]$, and $V_a/S_a$
(figures not shown here). The dominant influence of $[H_2O]$ on the $\gamma(N_2O_5)$ value was also
reported in a previous study in north China (Tham et al., 2018).
The $\varphi(ClNO_2)$ value ranged from 0.28 to 0.89 (mean, 0.56 ± 0.15), which was among the



highest values in the world (McDuffie et al., 2018b). The $\varphi(ClNO_2)$ value in this study exhibited
an obvious nonlinear relationship with the $[Cl^-]/[H_2O]$ ratio ($R^2 = 0.52$) (Fig. 4a), which is
consistent with previous laboratory results (Bertram and Thornton, 2009). However, current
parameterization of $\varphi(ClNO_2)$ based on $[Cl^-]/[H_2O]$ ($\varphi(ClNO_2)_{BT}$) tended to overestimate the
observed $\varphi(ClNO_2)$ value (Fig. 4b).
Here we give two explanations for the inconsistency between the $\varphi(ClNO_2)_{BT}$ and the field-
derived $\varphi(ClNO_2)$. First, the reactivity of chloride with $NO_2^+$ (i.e., $k_4/k_3$ in Eq. 8) was reduced
in ambient environments due to complicated issues of the mixing state, phase state, and activity
coefficient. As $\varphi(ClNO_2)$ is positively dependent upon $[Cl^-]$, a reduction in chloride reactivity
could decrease the $\varphi(ClNO_2)$ value in ambient particles. This explanation is supported by
previous studies of $\gamma(N_2O_5)$ (Morgan et al., 2015; McDuffie et al., 2018a), which showed that
when the enhancement effect of chloride on $\gamma(N_2O_5)$ was neglected, the parameterized $\gamma(N_2O_5)$
better matched the observed $\gamma(N_2O_5)$. The second explanation deals with other unknown factors
that reduce the $\varphi(ClNO_2)$ value. The parameterization $\varphi(ClNO_2)_{BT}$ only considered the
$[Cl^-]/[H_2O]$ ratio, not other aqueous species that could suppress $\varphi(ClNO_2)$, leading to the
overestimation of $\varphi(ClNO_2)_{BT}$ values.
Regarding the second explanation, we examined the possibility of sulfate and aerosol
organics competing with $[Cl^-]$ for the $NO_2^+$ intermediate (see Section 2.4 and Eq. (10)). The
statistical results show that aerosol organics could reduce $\varphi(ClNO_2)$ values ($R^2 = 0.41$; Fig. S7b),
but sulfate did not show such an influence ($R^2 = 0.08$; Fig. S7a). The latter result contrasts with
the finding of a recent laboratory study, which indicated that both sulfate and some organics
(e.g., carboxylate) suppress $ClNO_2$ formation (Staudt et al., 2019).
By incorporating the suppression effect of aerosol organics, we performed regressions of
$\varphi(ClNO_2)$ and obtained an improved parameterization of $\varphi(ClNO_2)$ (noted as $\varphi(ClNO_2)_{BT+Org}$)
that better matched the observed $\varphi(ClNO_2)$ (Fig. 4b). In Eq. (11), the factor 483 ($k_4/k_3$ in Eq. 9)
was adopted from (Bertram and Thornton, 2009), and the factor 235 ($k_4/k_5$ in Eq. 9) was derived
here by iterative algorithms to achieve the least-square errors between the observed and
parameterized $\varphi(ClNO_2)$ values. Given that $k_4/k_3 = 483$ and $k_4/k_5 = 235$, $k_5/k_3$ was calculated as



2.06, which suggests that the reaction rate constant of aerosol organics with $NO_2^+$ was twice
that of the $H_2O + NO_2^+$ reaction. A recent laboratory study (Staudt et al., 2019) derived $k_5/k_3 =$
3.7 for acetate, which is very similar to our results.
$$\varphi(ClNO_2)_{BT+Org} = \left(1 + \frac{[H_2O]}{483[Cl^-]} + \frac{[Org]}{235[Cl^-]}\right)^{-1} \tag{11}$$

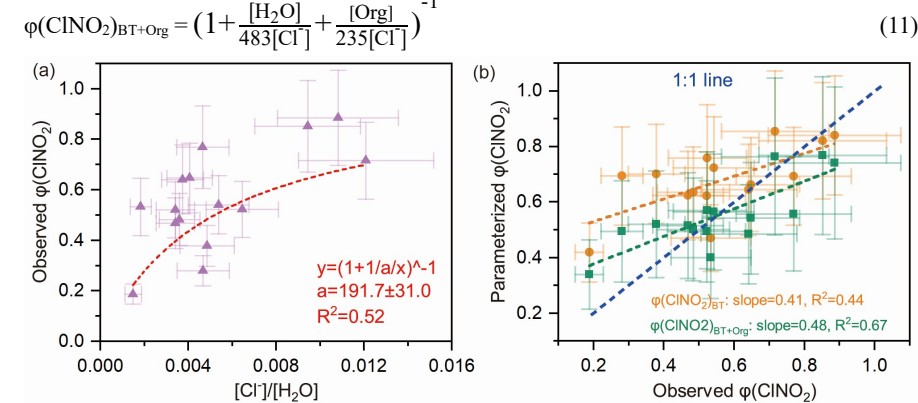

**Figure 4.** Influencing factors and parameterizations of $\varphi(ClNO_2)$. **(a)** Dependence of $\varphi(ClNO_2)$
on the $[Cl^-]/[H_2O]$ ratio. Dashed red line shows nonlinear fitting of $\varphi(ClNO_2)$; "a" represents
the $k_4/k_3$ in Eq. (8). **(b)** Comparison of parameterized $\varphi(ClNO_2)$ and observed $\varphi(ClNO_2)$, where
$\varphi(ClNO_2)_{BT}$ denotes the parameterization proposed by Bertram and Thornton (2009), and
$\varphi(ClNO_2)_{BT+Org}$ represents the revised parameterization used in this study (see Eq. (11)).

**3.4 Nocturnal $Cl_2$ formation**
**3.4.1 $Cl_2$ as a co-product of $ClNO_2$ from $N_2O_5$ uptake**
To elucidate the formation pathways of the elevated levels of $Cl_2$ observed during the night,
we investigated the correlations of $Cl_2$ with the $ClNO_2$, $HOCl$, and $SO_2$ and the diurnal
variations of these (Fig. 5a–5d). $Cl_2$ only exhibited a significant correlation with $ClNO_2$ (Fig
5a). The $Cl_2/ClNO_2$ ratios varied on different nights, which implies that differences exist in the
production efficiencies of $Cl_2$ relative to those of $ClNO_2$.

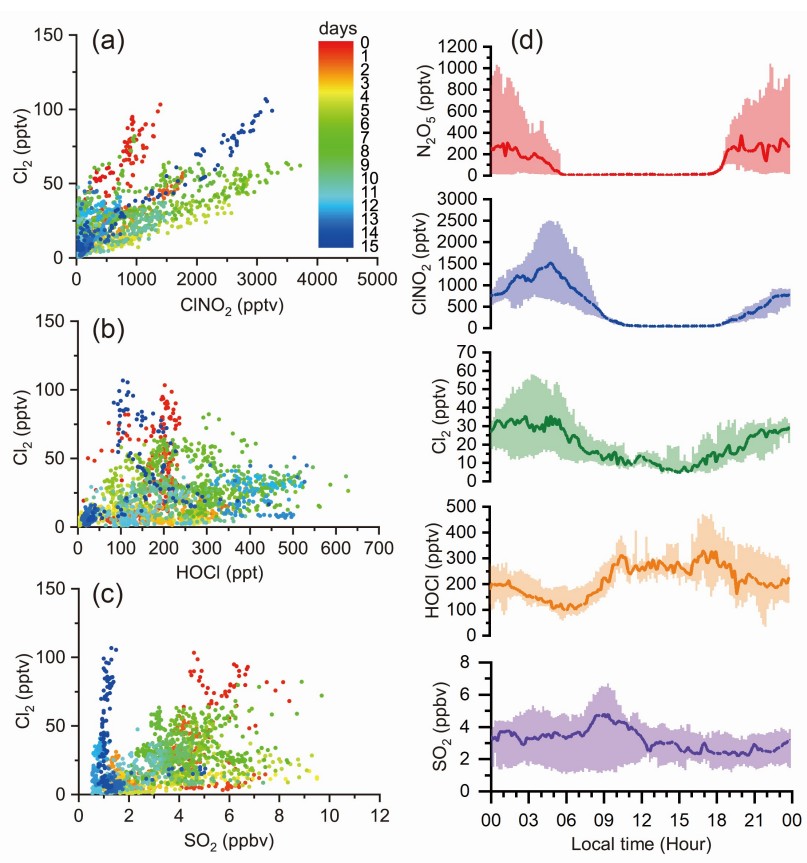


**Figure 5.** Correlations among $Cl_2$, $ClNO_2$, HOCl, and $SO_2$ and their diurnal profiles **(a)**, **(b)**,
and **(c)** show the correlations of $Cl_2$ with $ClNO_2$, HOCl, and $SO_2$ respectively, during the whole
campaign. Dots represent 10-min averaged values colored according to campaign days. **(d)**
exhibits the diurnal variation of $Cl_2$, $ClNO_2$, HOCl, and $SO_2$.

358

The current mainstream interpretation of the observed correlation of $ClNO_2$ and $Cl_2$ is that
$Cl_2$ is produced from $ClNO_2$ uptake (Ammann et al., 2013; Qiu et al., 2019; Wang et al., 2019;
Haskins et al., 2019). We provide evidence that this interpretation does not apply to
measurements from our site. We assessed the $ClNO_2$ uptake hypothesis by examining the
magnitude of $\gamma(ClNO_2)$ needed to explain the nocturnal increase in $Cl_2$ mixing ratios and the
dependence of $\gamma(ClNO_2)$ on its known influencing factors. Assuming a unity yield of $Cl_2$ from
$ClNO_2$ uptake, the increasing rate of $Cl_2$ mixing ratios was calculated with Eq. (12). Eq. (13),





which was derived by rearrangement of Eq. (12), was adopted to estimate $\gamma(ClNO_2)$ via the
observed $Cl_2$ and $ClNO_2$ levels.
$d[Cl_2]/dt = \dfrac{1}{4}c(ClNO_2)S_a\gamma(ClNO_2)[ClNO_2]$                   (12)
$\gamma(ClNO_2)_{obs} = \dfrac{4d[Cl_2]/dt}{c(ClNO_2)S_a[ClNO_2]}$              (13)
where $c(ClNO_2)$ is the mean molecular velocity of $ClNO_2$ (m/s), and $[ClNO_2]$ represents the
averaged ambient concentration of $ClNO_2$ in the cases of interest.
$\gamma(ClNO_2)_{obs}$ was estimated in the selected cases following criteria 1 and 2 in Section 3.3, and
a steady increase in $Cl_2$ mixing ratios was required. The resulting values of $\gamma(ClNO_2)_{obs}$ were
compiled according to the local time and are presented in box charts (Fig. 6a). Fig. 6a also
shows the potential factors influencing $\gamma(ClNO_2)$: $[Cl^-]$, $[H^+]$, and particle diameters ($D_p$,
derived from the ratio of wet $V_a$ to $S_a$). Here, $[Cl^-]$ and $D_p$ were regarded as the factors
influencing $\gamma(ClNO_2)$ because a previous field study had found positive correlations of $\gamma(ClNO_2)$
with $[Cl^-]$ and $D_p$ (Haskins et al., 2019). $[H^+]$ was considered because the previous laboratory
study proposed that $H^+$ was as a reactant in $Cl_2$ production (Roberts et al., 2008). Each box
represents the $\gamma(ClNO_2)$, $[Cl^-]$, $[H^+]$, or $D_p$ of 10-min resolutions derived on individual days.
For example, the box for 18:00–19:00 contains the $\gamma(ClNO_2)$ estimated at 18:00–19:00 on 11,
12, and 14 April (Fig. 6b–6d). Fig. 6b–6d displays the observed $Cl_2$ levels and the projected
trend of $Cl_2$ levels obtained by use of Eq. (12).

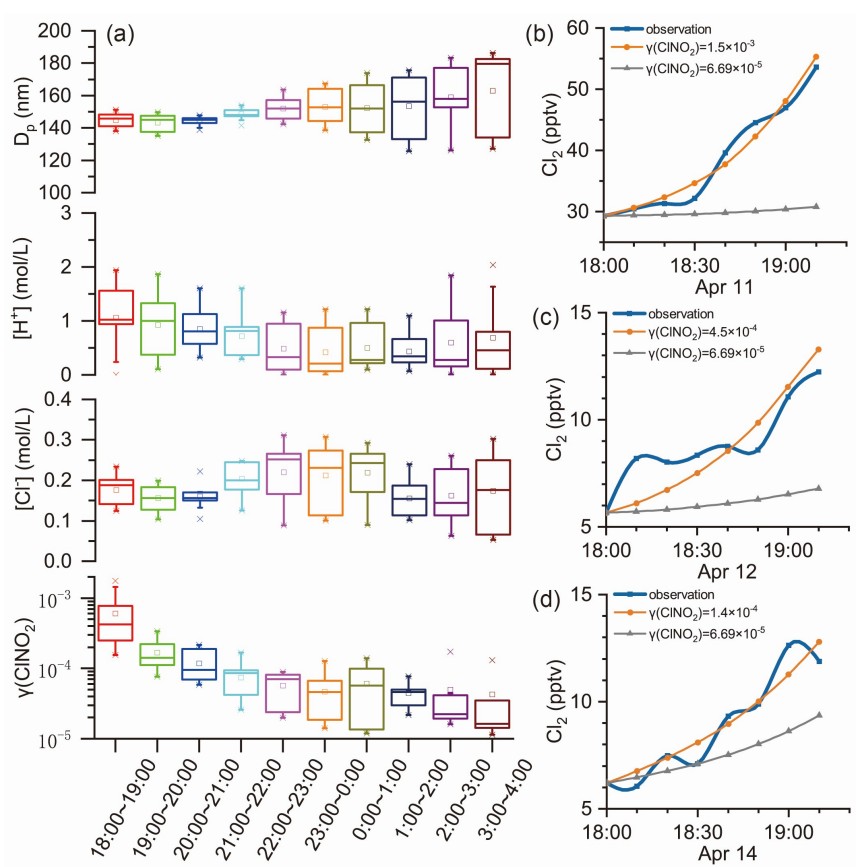


**Figure 6.** $\gamma(ClNO_2)$ estimated using field observation data. **(a)** $\gamma(ClNO_2)_{obs}$, $[Cl^-]$, $[H^+]$, and $D_p$

estimated at various nighttime periods. **(b)–(d)** Trends of increasing trends of $Cl_2$ mixing ratios

during the early evening hours on 11, 12, and 14 April, respectively. Orange and gray lines

represent the projected trend of $Cl_2$ mixing ratios using Eq. (12) with constant $\gamma(ClNO_2)$ values

and observed $ClNO_2$ levels.


As the increasing rate of $Cl_2$ concentrations $(d[Cl_2]/dt)$ did not change significantly during
the night (Fig. 5d), the $\gamma(ClNO_2)$ value was constrained by a sharp decreasing trend to
compensate for the increasing $ClNO_2$ levels after dusk (see Eq. 12). The highest $\gamma(ClNO_2)_{obs}$
value determined during the early evening hours (18:00–19:00) was similar to the laboratory-
derived $\gamma(ClNO_2)$ value on acidic salt films ($6 \times 10^{-3}$) (Roberts et al., 2008). However, the lowest
$\gamma(ClNO_2)_{obs}$ value estimated during later nighttime hours (22:00–04:00) was two orders of





magnitude lower ($10^{-5}$) and was comparable that observed during previous field observations
(Haskins et al., 2019). The large variations in the $\gamma(ClNO_2)$ value contrasted with the relatively
stable levels of $[Cl^-]$, $[H^+]$, and $D_p$ at various times of night, which is in opposition to the current
understanding of the relationship between the $\gamma(ClNO_2)$ and these factors. Therefore, the $ClNO_2$
uptake hypothesis cannot explain the nocturnal increase in $Cl_2$ mixing ratios that we observed
at our study site. We propose another hypothesis for the $ClNO_2$–$Cl_2$ correlation and suggest that
$Cl_2$ is a co-product of $ClNO_2$ produced from $N_2O_5$ uptake, in which $ClNO_2$ is not necessarily an
intermediate of $Cl_2$. However, further studies are needed to validate this hypothesis.
**3.4.2 Parameterizing $Cl_2$ formation from $N_2O_5$ uptake**
As the nighttime $Cl_2$, $ClNO_2$, and most nitrate ultimately originated from $N_2O_5$ uptake, we
can derive a relationship between $Cl_2$ and the $N_2O_5$ uptake. We assigned a production yield to
$Cl_2$ ($\varphi(Cl_2)$) from the $N_2O_5$ uptake analogous to the $ClNO_2$ yield and calculated this using Eq.

409  (14):

$$\varphi(Cl_2) = \frac{d[Cl_2]/dt}{k(N_2O_5)[N_2O_5]} \tag{14}$$
$\varphi(Cl_2)$ was estimated in the same cases in which $\gamma(N_2O_5)$ and $\varphi(ClNO_2)$ were derived, because
the availability of $\gamma(N_2O_5)$ was a prerequisite of deriving $\varphi(Cl_2)$. The estimated $\varphi(Cl_2)$ value
was 0.01–0.04 (Table S2). The dependences of $\varphi(Cl_2)$ on its potential influencing factors (i.e.,
$[Cl^-]$, $[H^+]$, and $D_p$) were examined. The results show that $\varphi(Cl_2)$ had positive correlations with
both $[Cl^-]$ ($R^2 = 0.74$) and $[H^+]$ ($R^2 = 0.75$) and that the data had a high $\varphi(Cl_2)$ region and a low
$\varphi(Cl_2)$ region (Fig. 7a, b). The low $\varphi(Cl_2)$ values were found in continental air masses with
relatively lower chloride concentrations, more alkaline ammonium, less acidic sulfate and
nitrate, and thus lower acidity (Fig. 7d), whereas the high $\varphi(Cl_2)$ values were associated with
marine air masses with higher loadings of aerosol chloride, less ammonium, and more acidic
compounds, and thus higher acidity (Fig. 7c). The higher acidity in the marine air masses may
be explained by the passage of this air mass over the industrialized cities in the YRD where
large concentrations of $SO_2$ and $NO_x$ are emitted. The dependences of the defined $\varphi(Cl_2)$ on
$[Cl^-]$ and $[H^+]$ indicate that nocturnal $Cl_2$ production requires the presence of highly acidic
chloride-rich particles and sufficient levels of $N_2O_5$.

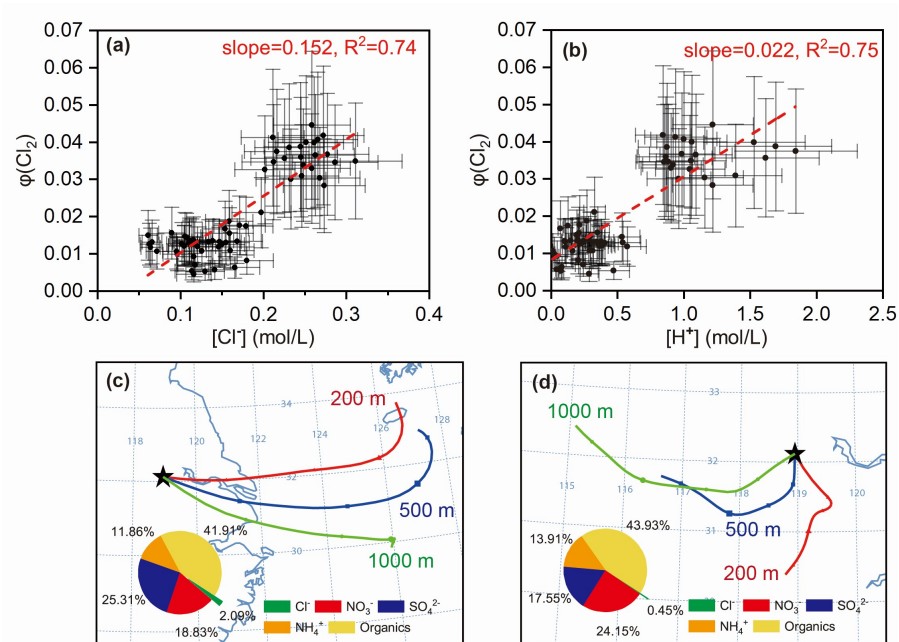

**Figure 7.** Estimated $\varphi(Cl_2)$ from $N_2O_5$ uptake and the factors influencing $\varphi(Cl_2)$ **(a)** and **(b)**

Dependencies of $\varphi(Cl_2)$ on $[Cl^-]$ and $[H^+]$ in selected cases. **(c)** and **(d)** are examples of high

$\varphi(Cl_2)$ values in marine air masses (e.g., 13 April) and low $\varphi(Cl_2)$ values in inland air masses

(e.g., 18 April) represented by 24-hour backward trajectories. Inserted pie charts show average

aerosol chemical compositions during 21:40 on 12 April to 00:40 on 13 April and from 22:20

to 23:40 17 April, respectively.

A parameterization scheme is derived based on the dependences of $\varphi(Cl_2)$ on $[Cl^-]$ and $[H^+]$

to predict the $Cl_2$ formation involving $N_2O_5$ heterogeneous chemistry. Mechanistically, it is

assumed that the nocturnal $Cl_2$ is produced from reactions involving $NO_2^+$. The production rates

of nitrate, $ClNO_2$, and $Cl_2$ from the loss of $NO_2^+$ are expressed in Eq. (15) through Eq. (17). The

loss rate of aerosol organics induced by $NO_2^+$ is expressed in Eq. (18) (noted as d[Org]/dt here).

$$d[NO_3^-]/dt = k_3[NO_2^+][H_2O] \tag{15}$$

$$d[ClNO_2]/dt = k_4[NO_2^+][Cl^-] \tag{16}$$

$$d[Cl_2]/dt = k_6[NO_2^+][Cl^-][H^+] \tag{17}$$

$$d[Org]/dt = k_5[NO_2^+][Org] \tag{18}$$





The symbol $k_6$ represents the rate constant of the reaction involving $NO_2^+$, $Cl^-$, and $H^+$. $\varphi(Cl_2)$
is obtained as follows, by assuming a steady state of the $NO_2^+$ intermediate (Bertram and
Thornton, 2009) (Eq. (19)).
$$\varphi(Cl_2)= \frac{\frac{d[Cl_2]}{dt}}{\frac{d[Cl_2]}{dt}+\frac{d[ClNO_2]}{dt}+\frac{d[NO_3^-]}{dt}+\frac{d[Org]}{dt}} = \frac{k_6[Cl^-][H^+]}{k_6[Cl^-][H^+]+k_4[Cl^-]+k_3[H_2O]+k_5[Org]} \qquad (19)$$
To remain consistent with the $\varphi(ClNO_2)$ parameterizations, the values 483 and 2.05 were
adopted for $k_4/k_3$ and $k_5/k_3$, respectively, while $k_6/k_3$ was estimated from the fitting of $\varphi(Cl_2)$
using Eq. (19) to achieve the least-squares errors between the observed and parameterized $\varphi(Cl_2)$
values. The parameterization of $\varphi(Cl_2)$ was then expressed as follows (Eq. (20)):
$$\varphi(Cl_2) = \frac{19.38[H^+][Cl^-]}{19.38[H^+][Cl^-]+483[Cl^-]+[H_2O]+2.05[Org]} \qquad (20)$$
where the units of $[H^+]$, $[Cl^-]$, and $[Org]$ are mol/L.
The previous $ClNO_2$ uptake method assumed a unity $Cl_2$ yield from $ClNO_2$ uptake, but no
such assumption is required in the new method for an explicit definition (Eq. 14) and
parameterization (Eq. 20) of the $\varphi(Cl_2)$. In addition, a quantitative relationship between $\varphi(Cl_2)$
and aerosol acidity is established, which was not given in the previous parameterization.

**4. Summary and conclusions**
This study reports the presence of significant levels of $ClNO_2$ and $Cl_2$ at a suburban site in
east China. A rapid increase in the $ClNO_2$ mixing ratios was found to occur after midnight due
to larger rates of $N_2O_5$ heterogeneous loss than in early nighttime hours, and a high $\varphi(ClNO_2)$
value was also responsible for the elevated $ClNO_2$ mixing ratios. Improved parameterization of
$\varphi(ClNO_2)$ was achieved by involving the suppression effect of aerosol organics. We suggest
that the observed nighttime $Cl_2$ was a co-produced with $ClNO_2$ from the heterogeneous $N_2O_5$
uptake on acidic aerosols that bear high concentrations of chloride, and we also propose a
parameterization for $\varphi(Cl_2)$. The combination of $\varphi(Cl_2)$, $\varphi(ClNO_2)$, and $\gamma(N_2O_5)$ can be used to
predict the nighttime formation of $Cl_2$ and $ClNO_2$ from $N_2O_5$ uptake and their effect on the next
day's atmospheric photochemistry.



*Acknowledgments.* This study was supported by the National Natural Science Foundation
of China (NSFC) project (grant number: 91544213 and D0512/41675145), and the Hong
Kong Research Grants Council (T24-504/17-N). The authors acknowledge helpful
opinions and discussions from Dr. Yee Jun Tham.
*Author contributions.* TW designed the research. WN and AD managed the sampling sites. MX,
XP, and WW performed the CIMS measurements. CY, ZX, PS, YL, YL, ZX provided other
data. MX and TW wrote the manuscript with comments from all co-authors.
*Competing interests.* The authors declare that they have no conflict of interest.
*Data availability.* To request the CIMS, $jNO_2$, and $NO_y$ data described in this study, please
contact the corresponding author (cetwang@polyu.edu.hk). Other datasets are available by
contacting Dr. Wei Nie (niewei@nju.edu.cn).

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
