# Peer review of "Significant production of CINO2 and possible source of Cl2 from N2O5 uptake at a"

_Atmospheric Chemistry and Physics, 2019_

## Referee Comment (RC1) · Anonymous Referee #1 · 4 Feb 2020

The manuscript "Significant production of ClNO2 and possible source of Cl2 from N2O5 uptake at a suburban site in eastern China" by Xia et al. presents a set of measurements of nitryl chloride (ClNO2) and molecular chlorine (Cl2) taken near the city of Nanjing, in Eastern China, in April 2018. The authors use this dataset, and related observations, to analyze the formation of ClNO2 and Cl2 and to draw conclusions about the underlying multiphase chemical mechanism.

The paper is well written and the data are presented in a clear and concise way. The analysis and the results are sound and the authors propose some novel ideas that will certainly be of great interest to the community. I only have a few, fairly minor, com-

ments, but overall I think this paper is suitable for publication in Atmospheric Chemistry and Physics.

General Comments ——————-

In Section 3.3, the authors discuss the calculation of the yield of ClNO2, comparing the "BT" parametrization by Bertram and Thornton (2009) with a new parametrization. Looking at figure 4, I am not sure I completely agree with the author's interpretation. The new parametrization proposed in this paper does indeed agree better with the observations for yields between 0.4 and 0.6; however I would argue that the agreement is worse than the BT parametrization at higher yields (around 0.8) and only slightly better at lower yields (below 0.4). Clearly, the relationship between the various parameters is more complicated than either parametrization assume, and perhaps this suggests that there are other parameters that are not currently taken into account which play a role. In any case, I suggest that the authors revise their statements in this section (and the related parts of the conclusions and the abstract) to be more accurate.

In Section 3.4, the authors propose a mechanism for the production of Cl2 during the night. The key point of the argument is that, for the observations to be consistent with each other, g(ClNO2) must decrease and there is not really a good explanation for why that would be the case. Although I agree with this logic, there may be other parameters that influence g(ClNO2) besides Cl-, H+ and Dp. In particular organics, which are mentioned as important for g(N2O5) in the previous section may inhibit the uptake of ClNO2 as well. Likewise, RH, other aerosol components, and perhaps even temperature, may have an effect. I appreciate that it is not possible to exhaust all possible parameters but I think the authors should expand their analysis a little bit here, to make a more robust case.

The authors propose that Cl2 formation is a co-product of ClNO2 when N2O5 is hydrolized on an acidic particle. I would like to see a bit more discussion of this potential mechanism. For ClNO2 the mechanism is quite straightforward: NO2+ reacts with Cl-

to form ClNO2. For Cl2 it does not seem so obvious to me how exactly NO2+ and Cl-interact to form Cl2. If the authors have a mechanism in mind please explain or add the relevant reference(s). Otherwise, if this is simply an hypothesis, then please state so clearly.

Minor Comments ――――――

Section 2.1: Are there other relevant parameters (e.g., NOx) that you can use to compare the two sampling sites?

Section 2.2: Can you please add the detection limits to the text? It would also be useful to see examples of spectra for N2O5, ClNO2, Cl2 and HOCl (these could go in the Supplementary Information).

Line 187: what about NO3 photolysis?

Section 3.2: It seems to me, from figure 3, that the levels of VOC also play a role, not just O3, RH and Temperature.

Lines 416-423: What about the outflow from Nanjing, which is west of the sampling site? I would think there are industrialized areas also on that part of the country not just between Nanjing and the ocean. Are SO2 and NOx very different in the two cases shown in figure 7? Can you please add some detail.

---

## Referee Comment (RC2) · Anonymous Referee #2 · 9 Feb 2020

General Comment

The paper entitled with "Significant production of ClNO2 and possible source of Cl2 from N2O5 uptake at a suburban site in eastern China" presented comprehensive observations of N2O5, ClNO2 and Cl2 as well as other supporting parameters at a regional site in Nanjing. The authors performed a detailed studies on the heterogeneous processes subjected to N2O5 uptake and the chlorine productions. Some insights are given on the multiphase chemistry production of Cl2. This study further extends the current exploration of the nighttime chemistry in China from North China Plain and Pearl River Delta to Yangtze River Delta which are certainly valuable to be published

in ACP. Nevertheless, I think the current analysis needs some further careful check especially for the Section 3.4 as suggested in the follows.

Specific Comment

1. Line 149 – 150. More details need to be given for the sentence "the permeation rate of Cl2 was quantified by chemical titration and ultraviolet spectrophotometry." How much Cl2 is generated for calibration and what is the accuracy?

2. Section 3.2. The high ClNO2 case is of high interest. It would be nice if the authors can try to analyze why the ClNO2 production become higher for plume 3 than plume1. The Cl- ion concentrations seem to be quite small and constant for the whole period.

3. Line 306-307. "The $\varphi$(ClNO2) value ranged from 0.28 to 0.89 (mean, 0.56 $\pm$ 0.15), which was among the highest values in the world (McDuffie et al., 2018b)." I suggest to delete "which was among the highest values in the world (McDuffie et al., 2018b)." The $\varphi$(ClNO2) is varied within 0-1 depending on the ratio of [Cl-]/[H2O], so I do not think the highest is meaningful.

4. Line 338. The equation 11 and corresponding text. I think the estimation and the use of [org] needs more discussion. If the reaction between org and NO2+ is the key to formulate the equation, then the org should be the part of water soluble organics. And I wonder why the reaction with acetate can be similar to the field observations presented herein. What are the major water soluble organics here in this study? And actually you have two adjustable parameters, one is k5 and the other is the exact [org].

5. Line 375-376. The Dp is derived from the ratio of the wet Va to Sa. As I understood, the the dry Dp is measured directly from SMPS instrument and the wet Dp can be estimated from empirical GF factor or measurements if available. It may be worth to check two kinds of Dp for your calculations, one is for the surface area concentrations when it is surface limited, and the other is for the volume concentrations when it is limited by volume bulk reactions. The calculation of the Gamma_ClNO2 may be influenced by

the choice of the different Dp. A slightly different equation is suggested for your test of the gamma_ClNO2.

K_het = 1 / (Dp/Dg + 4/(gamma_ClNO2*c_ClNO2)) * 3 * ALW/Dp

Dg, gas diffusion constant

ALW, aerosol liquid water content

6. Line 397 – 400. The ALW could be a variable to check for Cl2 production.

7. Section 3.4.1 and 3.4.2, if the essence of Cl2 production is from ClNO2(aq) + H+ + Cl- → Cl2(g) Both the production of ClNO2 uptake and N2O5 uptake which can generate ClNO2(aq) could be the explanation for the Cl2 production. The authors may then to quantify the ratio of these two channels from the observations. In addition, the HOCl channel can also be assessed.

8. Section 3.4.2, the analysis of phi(ClNO2) is only meaningful, if the authors can prove the N2O5 uptake is the major (i.e. >90%) production channel of the Cl2.

---

## Author Comment (AC1) · 20 Mar 2020

The manuscript "Significant production of ClNO2 and possible source of Cl2 from N2O5 uptake at a suburban site in eastern China" by Xia et al. presents a set of measurements of nitryl chloride (ClNO2) and molecular chlorine (Cl2) taken near the city of Nanjing, in Eastern China, in April 2018. The authors use this dataset, and related observations, to analyze the formation of ClNO2 and Cl2 and to draw conclusions about the underlying multiphase chemical mechanism. The paper is well written and the data are presented in a clear and concise way. The analysis and the results are sound and the authors propose some novel ideas that will certainly be of great interest to the

community. I only have a few, fairly minor, comments, but overall I think this paper is suitable for publication in Atmospheric Chemistry and Physics. Response: we appreciate the reviewer for the positive comments and helpful suggestions. Below is the response to each comment. The reviewers' comments are italicized followed by our responses and changes shown in blue and red, respectively. And the corrections are also marked as red color in the revised manuscript. Please note that the line numbers mentioned below refer to the original submission (line numbers in the revised version has changed). Please refer to the supplement of this author comment for better views such as font colors, subscript and so on.

General Comments In Section 3.3, the authors discuss the calculation of the yield of ClNO2, comparing the "BT" parametrization by Bertram and Thornton (2009) with a new parametrization. Looking at figure 4, I am not sure I completely agree with the author's interpretation. The new parametrization proposed in this paper does indeed agree better with the observations for yields between 0.4 and 0.6; however I would argue that the agreement is worse than the BT parametrization at higher yields (around 0.8) and only slightly better at lower yields (below 0.4). Clearly, the relationship between the various parameters is more complicated than either parametrization assume, and perhaps this suggests that there are other parameters that are not currently taken into account which play a role. In any case, I suggest that the authors revise their statements in this section (and the related parts of the conclusions and the abstract) to be more accurate. Response: we appreciate and agree your comment on our interpretation of the performance of the new parameterization at higher yields (0.75~1). We also agree that other unconstrained factors may influence $\varphi$(ClNO2). We have revised the relevant texts as below. Revision in the main text: Line 329-330 (section 3.3): The parameterized $\varphi$(ClNO2)BT+Org better matches the observed $\varphi$(ClNO2) at low to median yields (0~0.75) and the R2 and slope values in the linear regression are closer to 1 (Fig. 4b). However, the parameterized $\varphi$(ClNO2)BT+Org is smaller than the observed $\varphi$(ClNO2) at high yields (0.75~0.9), which may be attributable to other unconstrained factors in the parameterization, e.g., mixing state and phase state

issues.

In Section 3.4, the authors propose a mechanism for the production of Cl2 during the night. The key point of the argument is that, for the observations to be consistent with each other, g(ClNO2) must decrease and there is not really a good explanation for why that would be the case. Although I agree with this logic, there may be other parameters that influence g(ClNO2) besides Cl-, H+ and Dp. In particular organics, which are mentioned as important for g(N2O5) in the previous section may inhibit the uptake of ClNO2 as well. Likewise, RH, other aerosol components, and perhaps even temperature, may have an effect. I appreciate that it is not possible to exhaust all possible parameters but I think the authors should expand their analysis a little bit here, to make a more robust case. Response: we agree with the referee that other unconstrained factors, in addition to those examined, may influence the $\gamma$(ClNO2). We have now examined the dependence of $\gamma$(ClNO2) on RH, T, and other relevant aerosol components (e.g., NO3-, SO42-, NH4+, and aerosol organics). Results show no obvious dependence of $\gamma$(ClNO2) on those parameters. We have clarified this point as follows. Revision in the main text: Line 398-400 (section 3.4.1): In our study, the Dp was derived from the ratio of wet Va to Sa by assuming volume-limited uptake (Ammann et al., 2013). We also calculated Dp assuming surface-limited uptake and obtained similar Dp values to the volume-limited approach, and no correlation with $\gamma$(ClNO2)obs was indicated. Moreover, the $\gamma$(ClNO2)obs showed no obvious relationship with other factors such as T, RH, H2O, NO3-, SO42-, NH4+, and aerosol organics (figure not shown).

The authors propose that Cl2 formation is a co-product of ClNO2 when N2O5 is hydrolyzed on an acidic particle. I would like to see a bit more discussion of this potential mechanism. For ClNO2 the mechanism is quite straightforward: NO2+ reacts with Cl- to form ClNO2. For Cl2 it does not seem so obvious to me how exactly NO2+ and Cl interact to form Cl2. If the authors have a mechanism in mind please explain or add the relevant reference(s). Otherwise, if this is simply an hypothesis, then please state so clearly. Response: we agree it would make the contention much more convincing if we

can suggest the potential formation mechanism for Cl2 from reaction of Cl- and NO2+. Here is our proposed mechanism (see below figure). According to the hybrid orbital theory, the NO2+ ion has two non-bonded $\pi$ molecular orbitals due to participation of the d orbital of the central nitrogen atom (Baird et al., 1981). When Cl- attacks one of the $\pi$ molecular orbitals, ClNO2 is formed. In the same way, Cl- can attach to the other $\pi$ molecular orbitals of NO2+ and form a short-lived HNO2Cl2 intermediate in presence of H+. Then, HNO2Cl2 decomposes to produce HONO and Cl2.

Revision in the main text: Line 402-404 (section 3.4.1): The mechanism is depicted in Figure 7 and goes as follows. It is known that N2O5 hydrolysis on aerosol is responsible for the production of NO2+. According to the hybrid orbital theory, the NO2+ ion has two non-bonded $\pi$ molecular orbitals due to participation of the d orbital of the central nitrogen atom (Baird and Tayler, 1981). ClNO2 is formed via the nucleophilic addition of Cl- to one of the $\pi$ molecular orbitals of NO2+ (Figure 7a) (Taylor, 1990; Behnke et al., 1997). In the same way, we propose a side reaction that the second Cl- can attach to the other $\pi$ molecular orbital of NO2+ and form a short-lived HNO2Cl2 intermediate in presence of H+. It is proposed that the unstable HNO2Cl2 decomposes to produce Cl2 (and HONO) (Figure 7b). This mechanism can explain concurrent productions of Cl2 and ClNO2 from N2O5 hydrolysis but needs confirmation by additional laboratory and theoretical studies.

Minor Comments Section 2.1: Are there other relevant parameters (e.g., NOx) that you can use to compare the two sampling sites? Response: it is a pity that only simultaneous measurements of O3 were conducted at both sites.

Section 2.2: Can you please add the detection limits to the text? It would also be useful to see examples of spectra for N2O5, ClNO2, Cl2 and HOCl (these could go in the Supplementary Information). Response: agreed. We have added the detection limits of N2O5, ClNO2, and Cl2 in the main text, and an example of spectra in the supplementary information. Below is the revision. Revision in the main text: Line 153-154 (section 2.2): The detection limits ($3\sigma$) of N2O5, ClNO2, Cl2 were 7 pptv, 2 pptv,

and 5 pptv, respectively. Revision in the SI:

Figure S3. An example of the CIMS spectra taken at 18 April 01:00 LT.

Line 187: what about NO3 photolysis? Response: thanks for the reminder of NO3 photolysis, but in the present study, we focus on the nighttime chemistry of NO3. So, the k(NO3) here is the loss rate for nighttime, and photolysis is not included.

Section 3.2: It seems to me, from figure 3, that the levels of VOC also play a role, not just O3, RH and Temperature. Response: the role of VOCs had been included in the calculation the NO3 reactivity which is dependent on VOC levels. For example, In the plume 3, the NO3 reactivity due to VOCs decreased compared that in the plume 1. So, a larger proportion of NO3 was lost via N2O5 uptake in the plume 3, which promoted ClNO2 formation.

Lines 416-423: What about the outflow from Nanjing, which is west of the sampling site? I would think there are industrialized areas also on that part of the country not just between Nanjing and the ocean. Are SO2 and NOx very different in the two cases shown in figure 7? Can you please add some detail. Response: 1. this is a good point. We have examined backward trajectories for the whole observation period but did not identify air masses from urban Nanjing in the west. Please see the figure below. We have added the trajectories figure in the SI. Revision in the SI:

Figure S2. Daily backward trajectories arriving at the sampling sites during the field observation period.

2. The levels of NOx and SO2 are slightly higher in marine air compared with continental air in the two cases in figure 7 (see the table below). We have added this point in the main text.

Date NOx (ppb) SO2 (ppb) Note 13-Apr 13.1±3.1 3.9±0.1 Marine air passing YRD industry 18-Apr 11.5±0.6 3.3±0.3 Continental air

Revision in the main text: Line 420-422 (section 3.4.2): The average concentrations of

SO2 (3.9±0.1ppbv) and NOx (13.1±3.1 ppbv) in the marine air masses were higher than those (NOx: 11.5±0.6 ppbv, SO2: 3.3±0.3 ppbv) in the inland air masses.

References Ammann, M., Cox, R. A., Crowley, J. N., Jenkin, M. E., Mellouki, A., Rossi, M. J., Troe, J., and Wallington, T. J.: Evaluated kinetic and photochemical data for atmospheric chemistry: Volume VI – heterogeneous reactions with liquid substrates, Atmos. Chem. Phys., 13, 8045-8228, 10.5194/acp-13-8045-2013, 2013. Baird, N. C., and Taylor, K. F.: The stabilizing effect of d orbitals on the central nitrogen atom in nitrogen-oxygen molecules and ions, Chemical Physics Letters, 80, 83-86, https://doi.org/10.1016/0009-2614(81)80062-0, 1981. Behnke, W., George, C., Scheer, V., and Zetzsch, C.: Production and decay of ClNO2 from the reaction of gaseous N2O5 with NaCl solution: Bulk and aerosol experiments, Journal of Geophysical Research: Atmospheres, 102, 3795-3804, 1997. Brown, S. S., Dubé, W. P., Tham, Y. J., Zha, Q., Xue, L., Poon, S., Wang, Z., Blake, D. R., Tsui, W., and Parrish, D. D.: Nighttime chemistry at a high altitude site above Hong Kong, Journal of Geophysical Research: Atmospheres, 121, 2457-2475, 2016. Taylor, R., Electrophilic Aromatic Substitution, John Wiley, New York, 1990. Wang, Z., Wang, W., Tham, Y. J., Li, Q., Wang, H., Wen, L., Wang, X., and Wang, T.: Fast heterogeneous N2O5 uptake and ClNO2 production in power plant and industrial plumes observed in the nocturnal residual layer over the North China Plain, Atmospheric Chemistry and Physics, 17, 12361-12378, 2017.

Please also note the supplement to this comment:
https://www.atmos-chem-phys-discuss.net/acp-2019-1130/acp-2019-1130-AC1-supplement.pdf
* * *
(a)

(b)

**Fig. 1.** Figure 7. Proposed formation mechanisms of ClNO2 and Cl2 from N2O5 uptake. (a) production of ClNO2 from NO2+ and Cl-. (b) production of Cl2 from NO2+, Cl-, and H+.

[Figure]

**Fig. 2.** Figure S3. An example of the CIMS spectra taken at 18 April 01:00 LT.

[Figure]

Red: 200m height. Blue: 500 m height. Green: 1000 m height. Local time: 00:00.

**Fig. 3.** Figure S2. Daily backward trajectories arriving at the sampling sites during the field
observation period.

**Supplement:**

*The manuscript "Significant production of ClNO2 and possible source of Cl2 from N2O5 uptake at a suburban site in eastern China" by Xia et al. presents a set of measurements of nitryl chloride (ClNO2) and molecular chlorine (Cl2) taken near the city of Nanjing, in Eastern China, in April 2018. The authors use this dataset, and related observations, to analyze the formation of ClNO2 and Cl2 and to draw conclusions about the underlying multiphase chemical mechanism. The paper is well written and the data are presented in a clear and concise way. The analysis and the results are sound and the authors propose some novel ideas that will certainly be of great interest to the community. I only have a few, fairly minor, comments, but overall I think this paper is suitable for publication in Atmospheric Chemistry and Physics.*

Response: we appreciate the reviewer for the positive comments and helpful suggestions. Below is the response to each comment. The reviewers' comments are italicized followed by our responses and changes shown in blue and red, respectively. And the corrections are also marked as red color in the revised manuscript. Please note that the line numbers mentioned below refer to the original submission (line numbers in the revised version has changed).

*General Comments ——————-*
*In Section 3.3, the authors discuss the calculation of the yield of ClNO2, comparing the "BT" parametrization by Bertram and Thornton (2009) with a new parametrization. Looking at figure 4, I am not sure I completely agree with the author's interpretation. The new parametrization proposed in this paper does indeed agree better with the observations for yields between 0.4 and 0.6; however I would argue that the agreement is worse than the BT parametrization at higher yields (around 0.8) and only slightly better at lower yields (below 0.4). Clearly, the relationship between the various parameters is more complicated than either parametrization assume, and perhaps this suggests that there are other parameters that are not currently taken into account which play a role. In any case, I suggest that the authors revise their statements in this section (and the related parts of the conclusions and the abstract) to be more accurate.*

Response: we appreciate and agree your comment on our interpretation of the performance of the new parameterization at higher yields (0.75~1). We also agree that other unconstrained factors may influence φ(ClNO$_2$). We have revised the relevant texts as below.

Revision in the main text:
Line 329-330 (section 3.3): The parameterized φ(ClNO$_2$)$_{BT+Org}$ better matches the observed φ(ClNO$_2$) at low to median yields (0~0.75) and the R$^2$ and slope values in the linear regression are closer to 1 (Fig. 4b). However, the parameterized φ(ClNO$_2$)$_{BT+Org}$ is smaller than the observed φ(ClNO$_2$) at high yields (0.75~0.9), which may be attributable to other unconstrained factors in the parameterization, e.g., mixing state and phase state issues.

*In Section 3.4, the authors propose a mechanism for the production of Cl2 during the night.*

*The key point of the argument is that, for the observations to be consistent with each other, g(ClNO2) must decrease and there is not really a good explanation for why that would be the case. Although I agree with this logic, there may be other parameters that influence g(ClNO2) besides Cl-, H+ and Dp. In particular organics, which are mentioned as important for g(N2O5) in the previous section may inhibit the uptake of ClNO2 as well. Likewise, RH, other aerosol components, and perhaps even temperature, may have an effect. I appreciate that it is not possible to exhaust all possible parameters but I think the authors should expand their analysis a little bit here, to make a more robust case.*

Response: we agree with the referee that other unconstrained factors, in addition to those examined, may influence the $\gamma(ClNO_2)$. We have now examined the dependence of $\gamma(ClNO_2)$ on RH, T, and other relevant aerosol components (e.g., $NO_3^-$, $SO_4^{2-}$, $NH_4^+$, and aerosol organics). Results show no obvious dependence of $\gamma(ClNO_2)$ on those parameters. We have clarified this point as follows.

Revision in the main text:

Line 398-400 (section 3.4.1): In our study, the $D_p$ was derived from the ratio of wet $V_a$ to $S_a$ by assuming volume-limited uptake (Ammann et al., 2013). We also calculated $D_p$ assuming surface-limited uptake and obtained similar $D_p$ values to the volume-limited approach, and no correlation with $\gamma(ClNO_2)_{obs}$ was indicated. Moreover, the $\gamma(ClNO_2)_{obs}$ showed no obvious relationship with other factors such as T, RH, $H_2O$, $NO_3^-$, $SO_4^{2-}$, $NH_4^+$, and aerosol organics (figure not shown).

*The authors propose that Cl2 formation is a co-product of ClNO2 when N2O5 is hydrolyzed on an acidic particle. I would like to see a bit more discussion of this potential mechanism. For ClNO2 the mechanism is quite straightforward: NO2+ reacts with Cl- to form ClNO2. For Cl2 it does not seem so obvious to me how exactly NO2+ and Cl interact to form Cl2. If the authors have a mechanism in mind please explain or add the relevant reference(s). Otherwise, if this is simply an hypothesis, then please state so clearly.*

Response: we agree it would make the contention much more convincing if we can suggest the potential formation mechanism for $Cl_2$ from reaction of $Cl^-$ and $NO_2^+$. Here is our proposed mechanism (see below figure). According to the hybrid orbital theory, the $NO_2^+$ ion has two non-bonded $\pi$ molecular orbitals due to participation of the d orbital of the central nitrogen atom (Baird et al., 1981). When $Cl^-$ attacks one of the $\pi$ molecular orbitals, $ClNO_2$ is formed. In the same way, $Cl^-$ can attach to the other $\pi$ molecular orbitals of $NO_2^+$ and form a short-lived $HNO_2Cl_2$ intermediate in presence of $H^+$. Then, $HNO_2Cl_2$ decomposes to produce HONO and $Cl_2$.

[Figure]

(a)

(b)

Revision in the main text:

Line 402-404 (section 3.4.1): The mechanism is depicted in Figure 7 and goes as follows. It is known that $N_2O_5$ hydrolysis on aerosol is responsible for the production of $NO_2^+$. According to the hybrid orbital theory, the $NO_2^+$ ion has two non-bonded $\pi$ molecular orbitals due to participation of the d orbital of the central nitrogen atom (Baird and Tayler, 1981). $ClNO_2$ is formed via the nucleophilic addition of $Cl^-$ to one of the $\pi$ molecular orbitals of $NO_2^+$ (Figure 7a) (Taylor, 1990; Behnke et al., 1997). In the same way, we propose a side reaction that the second $Cl^-$ can attach to the other $\pi$ molecular orbital of $NO_2^+$ and form a short-lived $HNO_2Cl_2$ intermediate in presence of $H^+$. It is proposed that the unstable $HNO_2Cl_2$ decomposes to produce $Cl_2$ (and HONO) (Figure 7b). This mechanism can explain concurrent productions of $Cl_2$ and $ClNO_2$ from $N_2O_5$ hydrolysis but needs confirmation by additional laboratory and theoretical studies.

*Minor Comments ————*

*Section 2.1: Are there other relevant parameters (e.g., NOx) that you can use to compare the two sampling sites?*

Response: it is a pity that only simultaneous measurements of $O_3$ were conducted at both sites.

*Section 2.2: Can you please add the detection limits to the text? It would also be useful to see examples of spectra for N2O5, ClNO2, Cl2 and HOCl (these could go in the Supplementary Information).*

Response: agreed. We have added the detection limits of $N_2O_5$, $ClNO_2$, and $Cl_2$ in the main text, and an example of spectra in the supplementary information. Below is the revision.
Revision in the main text:
Line 153-154 (section 2.2): The detection limits ($3\sigma$) of $N_2O_5$, $ClNO_2$, $Cl_2$ were 7 pptv, 2 pptv,

and 5 pptv, respectively.
Revision in the SI:

[Figure]

Figure S3. An example of the CIMS spectra taken at 18 April 01:00 LT.

*Line 187: what about NO3 photolysis?*

Response: thanks for the reminder of $NO_3$ photolysis, but in the present study, we focus on the nighttime chemistry of $NO_3$. So, the $k(NO_3)$ here is the loss rate for nighttime, and photolysis is not included.

*Section 3.2: It seems to me, from figure 3, that the levels of VOC also play a role, not just O3, RH and Temperature.*

Response: the role of VOCs had been included in the calculation the $NO_3$ reactivity which is dependent on VOC levels. For example, In the plume 3, the $NO_3$ reactivity due to VOCs decreased compared that in the plume 1. So, a larger proportion of $NO_3$ was lost via $N_2O_5$ uptake in the plume 3, which promoted $ClNO_2$ formation.

*Lines 416-423: What about the outflow from Nanjing, which is west of the sampling site? I would think there are industrialized areas also on that part of the country not just between Nanjing and the ocean. Are SO2 and NOx very different in the two cases shown in figure 7? Can you please add some detail.*

Response: 1. this is a good point. We have examined backward trajectories for the whole observation period but did not identify air masses from urban Nanjing in the west. Please see the figure below. We have added the trajectories figure in the SI.
Revision in the SI:

[Figure]

Red: 200m height. Blue: 500 m height. Green: 1000 m height. Local time: 00:00.

Figure S2. Daily backward trajectories arriving at the sampling sites during the field observation period.

2. The levels of $NO_x$ and $SO_2$ are slightly higher in marine air compared with continental air in the two cases in figure 7 (see the table below). We have added this point in the main text.

| Date | $NO_x$ (ppb) | $SO_2$ (ppb) | Note |
|---|---|---|---|
| 13-Apr | 13.1±3.1 | 3.9±0.1 | Marine air passing YRD industry |
| 18-Apr | 11.5±0.6 | 3.3±0.3 | Continental air |

Revision in the main text:
Line 420-422 (section 3.4.2): The average concentrations of $SO_2$ (3.9±0.1ppbv) and $NO_x$ (13.1±3.1 ppbv) in the marine air masses were higher than those ($NO_x$: 11.5±0.6 ppbv, $SO_2$: 3.3±0.3 ppbv) in the inland air masses.

**References**

Ammann, M., Cox, R. A., Crowley, J. N., Jenkin, M. E., Mellouki, A., Rossi, M. J., Troe, J., and Wallington, T. J.: Evaluated kinetic and photochemical data for atmospheric chemistry: Volume VI – heterogeneous reactions with liquid substrates, Atmos. Chem. Phys., 13, 8045-8228, 10.5194/acp-13-8045-2013, 2013.

Baird, N. C., and Taylor, K. F.: The stabilizing effect of d orbitals on the central nitrogen atom in nitrogen-oxygen molecules and ions, Chemical Physics Letters, 80, 83-86, https://doi.org/10.1016/0009-2614(81)80062-0, 1981.

Behnke, W., George, C., Scheer, V., and Zetzsch, C.: Production and decay of $ClNO_2$ from the reaction of gaseous $N_2O_5$ with NaCl solution: Bulk and aerosol experiments, Journal of Geophysical Research: Atmospheres, 102, 3795-3804, 1997.

Brown, S. S., Dubé, W. P., Tham, Y. J., Zha, Q., Xue, L., Poon, S., Wang, Z., Blake, D. R., Tsui, W., and Parrish, D. D.: Nighttime chemistry at a high altitude site above Hong Kong, Journal of Geophysical Research: Atmospheres, 121, 2457-2475, 2016.

Taylor, R., Electrophilic Aromatic Substitution, John Wiley, New York, 1990.

Wang, Z., Wang, W., Tham, Y. J., Li, Q., Wang, H., Wen, L., Wang, X., and Wang, T.: Fast heterogeneous $N_2O_5$ uptake and $ClNO_2$ production in power plant and industrial plumes

observed in the nocturnal residual layer over the North China Plain, Atmospheric Chemistry and Physics, 17, 12361-12378, 2017.

---

## Author Comment (AC2) · 20 Mar 2020

General Comment The paper entitled with "Significant production of ClNO2 and possible source of Cl2 from N2O5 uptake at a suburban site in eastern China" presented comprehensive observations of N2O5, ClNO2 and Cl2 as well as other supporting parameters at a regional site in Nanjing. The authors performed a detailed studies on the heterogeneous processes subjected to N2O5 uptake and the chlorine productions. Some insights are given on the multiphase chemistry production of Cl2. This study further extends the current exploration of the nighttime chemistry in China from North China Plain and Pearl River Delta to Yangtze River Delta which are certainly valuable

to be published in ACP. Nevertheless, I think the current analysis needs some further careful check especially for the Section 3.4 as suggested in the follows.

Response: we appreciate the reviewer for the positive comments and helpful suggestions. Below is the response to each comment. The reviewers' comments are italicized, followed by our responses and changes shown in blue and red, respectively. And the corrections are also marked as red color in the revised manuscript. Please note that the line numbers mentioned below refer to the original submission (line numbers in the revised version has changed). Please refer to the supplement of this author comment for better views such as font color, subscripts, formulas, and so on.

Specific Comment 1. Line 149 – 150. More details need to be given for the sentence "the permeation rate of Cl2 was quantified by chemical titration and ultraviolet spectrophotometry." How much Cl2 is generated for calibration and what is the accuracy?

Response: we have added more details of Cl2 calibration. The permeation rate of Cl2 generated for calibration is 380 ± 20 ng/min. We have added further details of Cl2 calibration in the SI. Below is the revision.

Revision in the SI: Text S1.4. Details of Cl2 calibration The Cl2 standard was generated from a permeation tube heated to 40 ℃ and flushed by ultrapure nitrogen gas (20 sccm) and then diluted in humidified zero air (6 SLPM). During the field campaign, Cl2 from the permeation tube was introduced into a KI solution (2% wt) for 1 hour. The permeation rate of Cl2 (380 ± 20 ng/min) was calculated from the I3- concentration in the KI solution which was measured by ultraviolet–visible spectrophotometry at 351 nm.

2. Section 3.2. The high ClNO2 case is of high interest. It would be nice if the authors can try to analyze why the ClNO2 production become higher for plume 3 than plume1. The Cl- ion concentrations seem to be quite small and constant for the whole period.

Response: we did analyze the reason for the higher ClNO2 production in the plume

3. As shown in lines 273-277 and Fig. 3, larger proportion of NO3 was lost via N2O5 uptake in plume 3, which caused elevated ClNO2 production in plume 3 compared with plume 1. As the Cl- concentration was similar in plumes 1 and 3 (0.17 $\pm$ 0.02 and 0.19 $\pm$ 0.03 $\mu$g/m3, respectively), we did not attribute the higher ClNO2 in plume 3 to Cl- concentrations. Low chloride concentrations while high levels of ClNO2 were also observed in previous studies, where HCl condensation was proposed to replenish particulate chloride to sustain the ClNO2 production (Osthoff et al., 2008; Thornton et al., 2010). We have clarified this point in the revised manuscript.

Revision in the main text: Line 277-278 (section 3.2): Compared with the high levels of ClNO2 (up to 3.5 ppbv) on the night of 17 April, the concentration of Cl- was low and relatively constant ($\sim$0.1 ppbv) during that period. The low chloride but high ClNO2 levels were also observed in previous studies, and HCl partition was proposed to replenish particulate chloride to sustain the ClNO2 production (Osthoff et al., 2008; Thornton et al., 2010).

3. Line 306-307. "The $\varphi$(ClNO2) value ranged from 0.28 to 0.89 (mean, 0.56 _ 0.15), which was among the highest values in the world (McDuffie et al., 2018b)." I suggest to delete "which was among the highest values in the world (McDuffie et al., 2018b)." The $\varphi$(ClNO2) is varied within 0-1 depending on the ratio of [Cl-]/[H2O], so I do not think the highest is meaningful.

Response: we agree and will delete "which was among the highest values in the world".

4. Line 338. The equation 11 and corresponding text. I think the estimation and the use of [org] needs more discussion. If the reaction between org and NO2+ is the key to formulate the equation, then the org should be the part of water soluble organics. And I wonder why the reaction with acetate can be similar to the field observations presented herein. What are the major water soluble organics here in this study? And actually you have two adjustable parameters, one is k5 and the other is the exact [org].

Response: we agree that ideally the [org] here should be water-soluble organics. However, the water-soluble organics are not available in our study, and only total organics were measured on-site. So operationally we assume that organics are all water-soluble, similar to previous studies (McDuffie et al., 2018a; McDuffie et al., 2018b). The k5/k3 value derived here (2.06) happens to be similar to that of acetate. One possibility is that the aerosol organics have a weighted average k5/k3 value of 2.06. Due to limited observation species, we don't know the exact composition of water-soluble organics. k5 is the only adjustable parameter here. Since we assumed all observed aerosol organics to be water-soluble, the unknown water-soluble proportion of organics is factored in k5.

Revision in the main text: Line 331-334 (section 3.3) Here we assumed that the observed aerosol organics were all water-soluble and reactive toward NO2+, as previous studies did (McDuffie et al., 2018a; McDuffie et al., 2018b). The unknown water-soluble proportion of aerosol organics is factored in k5. Line 350-352 (section 3.3): A recent laboratory study (Staudt et al., 2019) derived k5/k3 = 3.7 for acetate, which happens to be very similar to our results.

5. Line 375-376. The Dp is derived from the ratio of the wet Va to Sa. As I understood, the the dry Dp is measured directly from SMPS instrument and the wet Dp can be estimated from empirical GF factor or measurements if available. It may be worth to check two kinds of Dp for your calculations, one is for the surface area concentrations when it is surface limited, and the other is for the volume concentrations when it is limited by volume bulk reactions. The calculation of the Gamma_ClNO2 may be influenced by the choice of the different Dp. A slightly different equation is suggested for your test of the gamma_ClNO2. K_het = 1 / (Dp/Dg + 4/(gamma_ClNO2*c_ClNO2)) * 3 * ALW/Dp Dg, gas diffusion constant ALW, aerosol liquid water content

Response: we calculated Dp (127.6±16.5 nm) based on the volume-limited uptake, we now also calculated Dp based on the surface-limited uptake which gives a similar result (122.6±26.7 nm). The $\gamma$(ClNO2) is independent of both the surface-limited Dp and volume-limited Dp. We appreciate the referee for suggesting a formula to test the

$\gamma$(ClNO2). We could not find the origin of this formula in literature, and it is not clear to us how we can use this formula to derive $\gamma$(ClNO2) from ambient measurements. In our study, we use below Eq. 13 to calculate $\gamma$(ClNO2), which does not involve Dp, thus the calculation of $\gamma$(ClNO2) was not influenced by the choice of Dp. $\gamma$(ClNO2)obs = "4d[Cl2]/dt" /"c(ClNO2)Sa[ClNO2]" (13)

Revision in the main text: Line 398-400 (section 3.4.1): In our study, the Dp was derived from the ratio of wet Va to Sa by assuming volume-limited uptake (Ammann et al., 2013). We also calculated Dp assuming surface-limited uptake and obtained similar Dp values to the volume-limited approach, and no correlation with $\gamma$(ClNO2)obs was indicated.

6. Line 397 – 400. The ALW could be a variable to check for Cl2 production.

Response: thanks for this suggestion. we have checked ALW in Cl2 production by investigating the dependence of $\gamma$(ClNO2) on [H2O]. However, no obvious correlation is found. We have clarified this point. Revision in the main text: Line 398-400 (section 3.4.1): ...Moreover, the $\gamma$(ClNO2)obs showed no obvious relationship with other factors such as T, RH, H2O, NO3-, SO42-, NH4+, and aerosol organics (figure not shown).

7. Section 3.4.1 and 3.4.2, if the essence of Cl2 production is from ClNO2(aq) + H+ + Cl- $\rightarrow$ Cl2(g) Both the production of ClNO2 uptake and N2O5 uptake which can generate ClNO2(aq) could be the explanation for the Cl2 production. The authors may then to quantify the ratio of these two channels from the observations. In addition, the HOCl channel can also be assessed.

Response: we think both ClNO2 uptake and direct N2O5 uptake can generate ClNO2(aq) and then produce Cl2. However, based on ambient measurements, we cannot separate the contribution of the two pathways, because an assumption to derive $\gamma$(ClNO2) was that Cl2 was all produced by ClNO2 uptake (Eq. 13, line 369). We will clarify in the revised draft that N2O5 uptake and ClNO2 uptake are indistinguishable in driving Cl2 production. Since HOCl and Cl2 were poorly correlated, we believe that the HOCl channel has minor contributions to Cl2 production at our site. Revision regarding this comment is made together with the comment 8.

8. Section 3.4.2, the analysis of phi(ClNO2) is only meaningful, if the authors can prove the N2O5 uptake is the major (i.e. >90%) production channel of the Cl2.

Response: we think that the referee meant $\varphi$(Cl2) in the above comment. In this paper, we attempt to demonstrate/argue that the three previously proposed reaction pathways could not explain the observed Cl2 productions at night at our site, and suggest an additional one. We summarize our views as follows. 1. For ClNO2→Cl2, in section 3.4.1, we showed that the $\gamma$(ClNO2) derived from the assumption that Cl2 was from ClNO2 uptake didn't have the expected relationships with Cl-, H+, and Dp, which are the known factors that influence the ClNO2 uptake. Thus, we argue that it could not be the main pathway for Cl2 production at our site. We note that for our proposed new Cl2 pathway: NO2+ + Cl- + H+, we do not rule out the ClNO2 uptake channel, but assume it can produce Cl2 via NO2+. 2. For HOCl→Cl2, we think it is a minor Cl2 production channel, because Cl2 was not correlated to HOCl but highly correlated with ClNO2 during most nights. The same logic was adopted in a previous study (Haskins et al., 2019) to support the view of insignificant role of HOCl. 3. For ClONO2→Cl2, ClONO2 were not measured in our study. According to a recent field study in north China, nocturnal ClONO2 levels was low (maximum~15 pptv). So, we assume that ClONO2→Cl2 was not a significant Cl2 formation pathway, given the $\gamma$(ClONO2) on the order of 10-3 (Burkholder et al., 2015).

Revision for comments 7 and 8 in the main text: Line 348-352 (section 3.4.1) Our result suggests that Cl2 was related to ClNO2, but the HOCl pathway (R5) and coal burning were of minor importance at our site. ClONO2 was not measured during our study. Recent field measurements at a rural site in northern China reported low ClONO2 levels at night (maximum ~ 15 pptv) (Breton et al., 2018). We believe that the ClONO2 levels at our site was also low, and production pathway (R6) was insignificant given low

$\gamma(ClONO2)$ ($\sim$10-3) (Haskins et al., 2019). At our site, the Cl2/ClNO2 ratios varied on different nights, which implies that differences exist in the production efficiencies of Cl2 relative to those of ClNO2.

Line 381-383 (section 3.4.1) For example, the box for 18:00–19:00 contains the $\gamma(ClNO2)$ estimated at 18:00–19:00 on 11, 12, and 14 April (Fig. 6b–6d, orange lines). Fig. 6b–6d displays the observed Cl2 levels (blue lines) and the projected trends of Cl2 levels from Eq. (12), where the grey lines adopted the highest $\gamma(ClNO2)$ value, 6.69$\times$10-5 observed in the field study of Haskins et al. (2019). During early evening hours (i.e., 18:00–19:00), the $\gamma(ClNO2)$ value derived in our study was one to two orders of magnitude higher than those in that study. This result implies that either ClNO2 uptake was much faster at our site or other pathways were involved in Cl2 production. We provide evidence below that the latter is likely the case.

Line 406-411 (section 3.4.1) We propose a new framework to estimate nighttime Cl2 production by treating Cl2, ClNO2, and most nitrate all ultimately originating from N2O5 uptake. We assign a production yield to Cl2 from the N2O5 uptake ($\varphi(Cl2)$) analogous to the ClNO2 yield and calculate this metric using Eq. (14): $\varphi(Cl2)$ = ("d" ãĂŰ"[Cl" ãĂŮ_"2" "]/dt" )/("k(" "N" _"2" "O" _"5" )["N" _"2" "O" _"5" ] ) (14) The above formulation does not rule out the production of Cl2 from the ClNO2 uptake, because such production, if any, is also a result of N2O5 uptake and has thus been incorporated in Eq. (14).

Reference: Ammann, M., Cox, R. A., Crowley, J. N., Jenkin, M. E., Mellouki, A., Rossi, M. J., Troe, J., and Wallington, T. J.: Evaluated kinetic and photochemical data for atmospheric chemistry: Volume VI – heterogeneous reactions with liquid substrates, Atmos. Chem. Phys., 13, 8045-8228, 10.5194/acp-13-8045-2013, 2013. Bertram, T., and Thornton, J.: Toward a general parameterization of N2O5 reactivity on aqueous particles: the competing effects of particle liquid water, nitrate and chloride, Atmospheric Chemistry and Physics, 9, 8351-8363, 2009. Breton, M. L., Hallquist, Å. M., Pathak, R. K., Simpson, D., Wang, Y., Johansson, J., Zheng, J., Yang, Y., Shang,

D., and Wang, H.: Chlorine oxidation of VOCs at a semi-rural site in Beijing: significant chlorine liberation from ClNO2 and subsequent gas-and particle-phase Cl–VOC production, Atmospheric Chemistry and Physics, 18, 13013-13030, 2018. Burkholder, J. B., Sander, S. P., Abbatt, J. P. D., Barker, J. R., Huie, R. E., Kolb, C. E., et al. (2015). Chemical kinetics and photochemical data for use in atmospheric studies: Evaluation number 18. Pasadena, CA: Jet Propulsion Laboratory, National Aeronautics and Space Administration. Gaston, C. J., and Thornton, J. A.: Reacto-diffusive length of N2O5 in aqueous sulfate-and chloride-containing aerosol particles, The Journal of Physical Chemistry A, 120, 1039-1045, 2016. Haskins, J. D., Lee, B. H., Lopez‐Hili- fiker, F. D., Peng, Q., Jaeglé, L., Reeves, J. M., Schroder, J. C., Campuzano‐Jost, P., Fibiger, D., and McDuffie, E. E.: Observational constraints on the formation of Cl2 from the reactive uptake of ClNO2 on aerosols in the polluted marine boundary layer, Journal of Geophysical Research: Atmospheres, 124, 8851-8869, 2019. McDuffie, E. E., Fibiger, D. L., Dubé, W. P., Lopez‐Hilfiker, F., Lee, B. H., Thornton, J. A., Shah, V., Jaeglé, L., Guo, H., and Weber, R. J.: Heterogeneous N2O5 uptake during winter: Aircraft measurements during the 2015 WINTER campaign and critical evaluation of current parameterizations, Journal of Geophysical Research: Atmospheres, 123, 4345-4372, 2018a. McDuffie, E. E., Fibiger, D. L., Dubé, W. P., Lopez Hilfiker, F., Lee, B. H., Jaeglé, L., Guo, H., Weber, R. J., Reeves, J. M., and Weinheimer, A. J.: ClNO2 yields from aircraft measurements during the 2015 WINTER campaign and critical evaluation of the current parameterization, Journal of Geophysical Research: Atmo- spheres, 123, 12,994-913,015, 2018b. Osthoff, H. D., Roberts, J. M., Ravishankara, A. R., Williams, E. J., Lerner, B. M., Sommariva, R., Bates, T. S., Coffman, D., Quinn, P. K., Dibb, J. E., Stark, H., Burkholder, J. B., Talukdar, R. K., Meagher, J., Fehsenfeld, F. C., and Brown, S. S.: High levels of nitryl chloride in the polluted subtropical marine boundary layer, Nature Geoscience, 1, 324-328, 10.1038/ngeo177, 2008. Roberts, J. M., Osthoff, H. D., Brown, S. S., and Ravishankara, A.: N2O5 oxidizes chloride to Cl2 in acidic atmospheric aerosol, Science, 321, 1059-1059, 2008. Thornton, J. A., Kercher, J. P., Riedel, T. P., Wagner, N. L., Cozic, J., Holloway, J. S., Dube, W. P.,

Wolfe, G. M., Quinn, P. K., Middlebrook, A. M., Alexander, B., and Brown, S. S.: A large atomic chlorine source inferred from mid-continental reactive nitrogen chemistry, Nature, 464, 271-274, 10.1038/nature08905, 2010.

Please also note the supplement to this comment:
https://www.atmos-chem-phys-discuss.net/acp-2019-1130/acp-2019-1130-AC2-supplement.pdf

**Supplement:**

*General Comment*

*The paper entitled with "Significant production of ClNO2 and possible source of Cl2 from N2O5 uptake at a suburban site in eastern China" presented comprehensive observations of N2O5, ClNO2 and Cl2 as well as other supporting parameters at a regional site in Nanjing. The authors performed a detailed studies on the heterogeneous processes subjected to N2O5 uptake and the chlorine productions. Some insights are given on the multiphase chemistry production of Cl2. This study further extends the current exploration of the nighttime chemistry in China from North China Plain and Pearl River Delta to Yangtze River Delta which are certainly valuable to be published in ACP. Nevertheless, I think the current analysis needs some further careful check especially for the Section 3.4 as suggested in the follows.*

Response: we appreciate the reviewer for the positive comments and helpful suggestions. Below is the response to each comment. The reviewers' comments are italicized, followed by our responses and changes shown in blue and red, respectively. And the corrections are also marked as red color in the revised manuscript. Please note that the line numbers mentioned below refer to the original submission (line numbers in the revised version has changed).

Specific Comment

1. Line 149 – 150. More details need to be given for the sentence "the permeation rate of Cl2 was quantified by chemical titration and ultraviolet spectrophotometry." How much Cl2 is generated for calibration and what is the accuracy?

Response: we have added more details of Cl$_2$ calibration. The permeation rate of Cl$_2$ generated for calibration is 380 ± 20 ng/min. We have added further details of Cl$_2$ calibration in the SI. Below is the revision.

Revision in the SI:

Text S1.4. Details of Cl$_2$ calibration

The Cl$_2$ standard was generated from a permeation tube heated to 40 °C and flushed by ultrapure nitrogen gas (20 sccm) and then diluted in humidified zero air (6 SLPM). During the field campaign, Cl$_2$ from the permeation tube was introduced into a KI solution (2% wt) for 1 hour. The permeation rate of Cl$_2$ (380 ± 20 ng/min) was calculated from the I$_3^-$ concentration in the KI solution which was measured by ultraviolet–visible spectrophotometry at 351 nm.

2. Section 3.2. The high ClNO2 case is of high interest. It would be nice if the authors can try to analyze why the ClNO2 production become higher for plume 3 than plume1. The Cl- ion concentrations seem to be quite small and constant for the whole period.

Response: we did analyze the reason for the higher ClNO$_2$ production in the plume 3. As shown in lines 273-277 and Fig. 3, larger proportion of NO$_3$ was lost via N$_2$O$_5$ uptake in plume 3, which caused elevated ClNO$_2$ production in plume 3 compared with plume 1. As the Cl$^-$ concentration was similar in plumes 1 and 3 (0.17 ± 0.02 and 0.19 ± 0.03 μg/m$^3$, respectively), we did not attribute the higher ClNO$_2$ in plume 3 to Cl$^-$ concentrations.

Low chloride concentrations while high levels of ClNO$_2$ were also observed in previous studies, where HCl condensation was proposed to replenish particulate chloride to sustain the ClNO$_2$ production (Osthoff et al., 2008; Thornton et al., 2010). We have clarified this point in the revised manuscript.

Revision in the main text:

Line 277-278 (section 3.2): Compared with the high levels of ClNO$_2$ (up to 3.5 ppbv) on the night of 17 April, the concentration of Cl$^-$ was low and relatively constant (~0.1 ppbv) during that period. The low chloride but high ClNO$_2$ levels were also observed in previous studies, and HCl partition was proposed to replenish particulate chloride to sustain the ClNO$_2$ production (Osthoff et al., 2008; Thornton et al., 2010).

3. Line 306-307. "The φ(ClNO2) value ranged from 0.28 to 0.89 (mean, 0.56 _ 0.15), which was among the highest values in the world (McDuffie et al., 2018b)." I suggest to delete "which was among the highest values in the world (McDuffie et al., 2018b)." The φ(ClNO2) is varied within 0-1 depending on the ratio of [Cl-]/[H2O], so I do not think the highest is meaningful.

Response: we agree and will delete "which was among the highest values in the world".

4. Line 338. The equation 11 and corresponding text. I think the estimation and the use of [org] needs more discussion. If the reaction between org and NO2+ is the key to formulate the equation, then the org should be the part of water soluble organics. And I wonder why the reaction with acetate can be similar to the field observations presented herein. What are the major water soluble organics here in this study? And actually you have two adjustable parameters, one is k5 and the other is the exact [org].

Response: we agree that ideally the [org] here should be water-soluble organics. However, the water-soluble organics are not available in our study, and only total organics were measured on-site. So operationally we assume that organics are all water-soluble, similar to previous studies (McDuffie et al., 2018a; McDuffie et al., 2018b).

The $k_5/k_3$ value derived here (2.06) happens to be similar to that of acetate. One possibility is that the aerosol organics have a weighted average $k_5/k_3$ value of 2.06. Due to limited observation species, we don't know the exact composition of water-soluble organics.

$k_5$ is the only adjustable parameter here. Since we assumed all observed aerosol organics to be water-soluble, the unknown water-soluble proportion of organics is factored in k$_5$.

Revision in the main text:

Line 331-334 (section 3.3) Here we assumed that the observed aerosol organics were all water-soluble and reactive toward NO$_2^+$, as previous studies did (McDuffie et al., 2018a; McDuffie et al., 2018b). The unknown water-soluble proportion of aerosol organics is factored in $k_5$.

Line 350-352 (section 3.3): A recent laboratory study (Staudt et al., 2019) derived $k_5/k_3 = 3.7$ for acetate, which happens to be very similar to our results.

5. Line 375-376. The Dp is derived from the ratio of the wet Va to Sa. As I understood, the the dry Dp is measured directly from SMPS instrument and the wet Dp can be estimated from empirical GF factor or measurements if available. It may be worth to check two kinds of Dp for your calculations, one is for the surface area concentrations when it is surface limited, and the other is for the volume concentrations when it is limited by volume bulk reactions. The

calculation of the Gamma_ClNO2 may be influenced by the choice of the different Dp. A slightly different equation is suggested for your test of the gamma_ClNO2.

K_het = 1 / (Dp/Dg + 4/(gamma_ClNO2*c_ClNO2)) * 3 * ALW/Dp

Dg, gas diffusion constant

ALW, aerosol liquid water content

Response: we calculated $D_p$ (127.6±16.5 nm) based on the volume-limited uptake, we now also calculated $D_p$ based on the surface-limited uptake which gives a similar result (122.6±26.7 nm). The $\gamma(ClNO_2)$ is independent of both the surface-limited $D_p$ and volume-limited $D_p$.

We appreciate the referee for suggesting a formula to test the $\gamma(ClNO_2)$. We could not find the origin of this formula in literature, and it is not clear to us how we can use this formula to derive $\gamma(ClNO_2)$ from ambient measurements. In our study, we use below Eq. 13 to calculate $\gamma(ClNO_2)$, which does not involve $D_p$, thus the calculation of $\gamma(ClNO_2)$ was not influenced by the choice of $D_p$.

$$\gamma(ClNO_2)_{obs} = \frac{4d[Cl_2]/dt}{c(ClNO_2)S_a[ClNO_2]} \hspace{2cm} (13)$$

Revision in the main text:

Line 398-400 (section 3.4.1): In our study, the $D_p$ was derived from the ratio of wet $V_a$ to $S_a$ by assuming volume-limited uptake (Ammann et al., 2013). We also calculated $D_p$ assuming surface-limited uptake and obtained similar $D_p$ values to the volume-limited approach, and no correlation with $\gamma(ClNO_2)_{obs}$ was indicated.

6. Line 397 – 400. The ALW could be a variable to check for Cl2 production.

Response: thanks for this suggestion. we have checked ALW in $Cl_2$ production by investigating the dependence of $\gamma(ClNO_2)$ on $[H_2O]$. However, no obvious correlation is found. We have clarified this point.

Revision in the main text:

Line 398-400 (section 3.4.1): …Moreover, the $\gamma(ClNO_2)_{obs}$ showed no obvious relationship with other factors such as T, RH, $H_2O$, $NO_3^-$, $SO_4^{2-}$, $NH_4^+$, and aerosol organics (figure not shown).

7. Section 3.4.1 and 3.4.2, if the essence of Cl2 production is from ClNO2(aq) + H+ + Cl- → Cl2(g) Both the production of ClNO2 uptake and N2O5 uptake which can generate ClNO2(aq) could be the explanation for the Cl2 production. The authors may then to quantify the ratio of these two channels from the observations. In addition, the HOCl channel can also be assessed.

Response: we think both $ClNO_2$ uptake and direct $N_2O_5$ uptake can generate $ClNO_2(aq)$ and then produce $Cl_2$. However, based on ambient measurements, we cannot separate the contribution of the two pathways, because an assumption to derive $\gamma(ClNO_2)$ was that $Cl_2$ was all produced by $ClNO_2$ uptake (Eq. 13, line 369). We will clarify in the revised draft that $N_2O_5$ uptake and $ClNO_2$ uptake are indistinguishable in driving $Cl_2$ production.

Since HOCl and $Cl_2$ were poorly correlated, we believe that the HOCl channel has minor contributions to $Cl_2$ production at our site.

Revision regarding this comment is made together with the comment 8.

8. Section 3.4.2, the analysis of phi(ClNO2) is only meaningful, if the authors can prove the

N2O5 uptake is the major (i.e. >90%) production channel of the Cl2.

Response: we think that the referee meant $\varphi(Cl_2)$ in the above comment. In this paper, we attempt to demonstrate/argue that the three previously proposed reaction pathways could not explain the observed $Cl_2$ productions at night at our site, and suggest an additional one. We summarize our views as follows.

1. For $ClNO_2 \rightarrow Cl_2$, in section 3.4.1, we showed that the $\gamma(ClNO_2)$ derived from the assumption that $Cl_2$ was from $ClNO_2$ uptake didn't have the expected relationships with $Cl^-$, $H^+$, and $D_p$, which are the known factors that influence the $ClNO_2$ uptake. Thus, we argue that it could not be the main pathway for $Cl_2$ production at our site. We note that for our proposed new $Cl_2$ pathway: $NO_2^+ + Cl^- + H^+$, we do not rule out the $ClNO_2$ uptake channel, but assume it can produce $Cl_2$ via $NO_2^+$.

2. For $HOCl \rightarrow Cl_2$, we think it is a minor $Cl_2$ production channel, because $Cl_2$ was not correlated to HOCl but highly correlated with $ClNO_2$ during most nights. The same logic was adopted in a previous study (Haskins et al., 2019) to support the view of insignificant role of HOCl.

3. For $ClONO_2 \rightarrow Cl_2$, $ClONO_2$ were not measured in our study. According to a recent field study in north China, nocturnal $ClONO_2$ levels was low (maximum~15 pptv). So, we assume that $ClONO_2 \rightarrow Cl_2$ was not a significant $Cl_2$ formation pathway, given the $\gamma(ClONO_2)$ on the order of $10^{-3}$ (Burkholder et al., 2015).

Revision for comments 7 and 8 in the main text:

Line 348-352 (section 3.4.1)

Our result suggests that $Cl_2$ was related to $ClNO_2$, but the HOCl pathway (R5) and coal burning were of minor importance at our site. $ClONO_2$ was not measured during our study. Recent field measurements at a rural site in northern China reported low $ClONO_2$ levels at night (maximum ~ 15 pptv) (Breton et al., 2018). We believe that the $ClONO_2$ levels at our site was also low, and production pathway (R6) was insignificant given low $\gamma(ClONO_2)$ ($\sim10^{-3}$) (Haskins et al., 2019). At our site, the $Cl_2/ClNO_2$ ratios varied on different nights, which implies that differences exist in the production efficiencies of $Cl_2$ relative to those of $ClNO_2$.

Line 381-383 (section 3.4.1)

For example, the box for 18:00–19:00 contains the $\gamma(ClNO_2)$ estimated at 18:00–19:00 on 11, 12, and 14 April (Fig. 6b–6d, orange lines). Fig. 6b–6d displays the observed $Cl_2$ levels (blue lines) and the projected trends of $Cl_2$ levels from Eq. (12), where the grey lines adopted the highest $\gamma(ClNO_2)$ value, $6.69\times10^{-5}$ observed in the field study of Haskins et al. (2019). During early evening hours (i.e., 18:00–19:00), the $\gamma(ClNO_2)$ value derived in our study was one to two orders of magnitude higher than those in that study. This result implies that either $ClNO_2$ uptake was much faster at our site or other pathways were involved in $Cl_2$ production. We provide evidence below that the latter is likely the case.

Line 406-411 (section 3.4.1)

We propose a new framework to estimate nighttime $Cl_2$ production by treating $Cl_2$, $ClNO_2$,

and most nitrate all ultimately originating from $N_2O_5$ uptake. We assign a production yield to $Cl_2$ from the $N_2O_5$ uptake ($\varphi(Cl_2)$) analogous to the $ClNO_2$ yield and calculate this metric using Eq. (14):

$$\varphi(Cl_2) = \frac{d[Cl_2]/dt}{k(N_2O_5)[N_2O_5]} \tag{14}$$

The above formulation does not rule out the production of $Cl_2$ from the $ClNO_2$ uptake, because such production, if any, is also a result of $N_2O_5$ uptake and has thus been incorporated in Eq. (14).

**Reference:**

Ammann, M., Cox, R. A., Crowley, J. N., Jenkin, M. E., Mellouki, A., Rossi, M. J., Troe, J., and Wallington, T. J.: Evaluated kinetic and photochemical data for atmospheric chemistry: Volume VI – heterogeneous reactions with liquid substrates, Atmos. Chem. Phys., 13, 8045-8228, 10.5194/acp-13-8045-2013, 2013.

Bertram, T., and Thornton, J.: Toward a general parameterization of $N_2O_5$ reactivity on aqueous particles: the competing effects of particle liquid water, nitrate and chloride, Atmospheric Chemistry and Physics, 9, 8351-8363, 2009.

Breton, M. L., Hallquist, Å. M., Pathak, R. K., Simpson, D., Wang, Y., Johansson, J., Zheng, J., Yang, Y., Shang, D., and Wang, H.: Chlorine oxidation of VOCs at a semi-rural site in Beijing: significant chlorine liberation from $ClNO_2$ and subsequent gas-and particle-phase Cl–VOC production, Atmospheric Chemistry and Physics, 18, 13013-13030, 2018.

Burkholder, J. B., Sander, S. P., Abbatt, J. P. D., Barker, J. R., Huie, R. E., Kolb, C. E., et al. (2015). Chemical kinetics and photochemical data for use in atmospheric studies: Evaluation number 18. Pasadena, CA: Jet Propulsion Laboratory, National Aeronautics and Space Administration.

Gaston, C. J., and Thornton, J. A.: Reacto-diffusive length of $N_2O_5$ in aqueous sulfate-and chloride-containing aerosol particles, The Journal of Physical Chemistry A, 120, 1039-1045, 2016.

Haskins, J. D., Lee, B. H., Lopez‐Hilifiker, F. D., Peng, Q., Jaeglé, L., Reeves, J. M., Schroder, J. C., Campuzano‐Jost, P., Fibiger, D., and McDuffie, E. E.: Observational constraints on the formation of $Cl_2$ from the reactive uptake of $ClNO_2$ on aerosols in the polluted marine boundary layer, Journal of Geophysical Research: Atmospheres, 124, 8851-8869, 2019.

McDuffie, E. E., Fibiger, D. L., Dubé, W. P., Lopez‐Hilfiker, F., Lee, B. H., Thornton, J. A., Shah, V., Jaeglé, L., Guo, H., and Weber, R. J.: Heterogeneous $N_2O_5$ uptake during winter: Aircraft measurements during the 2015 WINTER campaign and critical evaluation of current parameterizations, Journal of Geophysical Research: Atmospheres, 123, 4345-4372, 2018a.

McDuffie, E. E., Fibiger, D. L., Dubé, W. P., Lopez Hilfiker, F., Lee, B. H., Jaeglé, L., Guo, H., Weber, R. J., Reeves, J. M., and Weinheimer, A. J.: $ClNO_2$ yields from aircraft measurements during the 2015 WINTER campaign and critical evaluation of the current parameterization, Journal of Geophysical Research: Atmospheres, 123, 12,994-913,015, 2018b.

Osthoff, H. D., Roberts, J. M., Ravishankara, A. R., Williams, E. J., Lerner, B. M., Sommariva, R., Bates, T. S., Coffman, D., Quinn, P. K., Dibb, J. E., Stark, H., Burkholder, J. B., Talukdar,

R. K., Meagher, J., Fehsenfeld, F. C., and Brown, S. S.: High levels of nitryl chloride in the polluted subtropical marine boundary layer, Nature Geoscience, 1, 324-328, 10.1038/ngeo177, 2008.

Roberts, J. M., Osthoff, H. D., Brown, S. S., and Ravishankara, A.: $N_2O_5$ oxidizes chloride to $Cl_2$ in acidic atmospheric aerosol, Science, 321, 1059-1059, 2008.

Thornton, J. A., Kercher, J. P., Riedel, T. P., Wagner, N. L., Cozic, J., Holloway, J. S., Dube, W. P., Wolfe, G. M., Quinn, P. K., Middlebrook, A. M., Alexander, B., and Brown, S. S.: A large atomic chlorine source inferred from mid-continental reactive nitrogen chemistry, Nature, 464, 271-274, 10.1038/nature08905, 2010.